# First-order photon condensation in magnetic cavities: A two-leg ladder model

Zeno Bacciconi[1,2⋆], Gian M. Andolina[3], Titas Chanda[1], Giuliano Chiriacò[1,2], Marco Schirò[3] and Marcello Dalmonte[1,2†]

**1** The Abdus Salam International Centre for Theoretical Physics (ICTP),
Strada Costiera 11, 34151 Trieste, Italy
**2** SISSA — International School of Advanced Studies,
via Bonomea 265, 34136 Trieste, Italy
**3** JEIP, USR 3573 CNRS, Collège de France, PSL Research University,
11 Place Marcelin Berthelot, F-75321 Paris, France

⋆ zbaccico@sissa.it , † mdalmont@ictp.it

## Abstract

We consider a model of free fermions in a ladder geometry coupled to a non-uniform cavity mode via Peierls substitution. Since the cavity mode generates a magnetic field, no-go theorems on spontaneous photon condensation do not apply, and we indeed observe a phase transition to a photon condensed phase characterized by finite circulating currents, alternatively referred to as the equilibrium superradiant phase. We consider both square and triangular ladder geometries, and characterize the transition by studying the energy structure of the system, light-matter entanglement, the properties of the photon mode, and chiral currents. The transition is of first order and corresponds to a sudden change in the fermionic band structure as well as the number of its Fermi points. Thanks to the quasi-one dimensional geometry we scrutinize the accuracy of (mean field) cavity-matter decoupling against large scale density-matrix renormalization group simulations. We find that light-matter entanglement is essential for capturing corrections to matter properties at finite sizes and for the description of the correct photon state. The latter remains Gaussian in the the thermodynamic limit both in the normal and photon condensed phases.

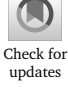

# 1 Introduction

One of the aims of the paradigm of cavity control is to modify the properties of quantum materials using cavity embedding [1–3]. In strong coupling regimes, vacuum effects [4] can modify the properties of the material even without external illumination, e.g., by affecting magneto-transport of a two dimensional (2D) material [5] or suppressing topological protection of the integer quantum Hall effect [6]. Recently, coupling to a cavity mode has been demonstrated to affect the critical temperature and the phase transition in charge-density wave systems [7]. Theoretical proposals have focused on the possibility of controlling electronic instabilities and ordered phases by quantum fluctuations of the cavity field, including superconductivity [8–10] and ferro-electricity [11, 12], or even inducing phase transitions in both light and matter degrees of freedom by onset of the so called *superradiant* phase where the ground state has a macroscopic number of coherent photons, hence photon condensed phase. The equilibrium superradiant phase transition, originally introduced in the context of the Dicke model [13–15] describing an ensemble of two-level atoms collectively coupled to a common cavity mode, has been recently discussed for electronic systems coupled to single-mode cavity [16–22].

A proper description of photon condensation requires a gauge invariant framework for the light-matter interaction, an issue which poses key theoretical challenges for truncated models which only retain a subset of degrees of freedom. In the ultrastrong coupling regime [23], where the light-matter coupling is comparable to the transition energies of the atoms, this truncation could lead to violations of gauge-invariance [24–26], thus questioning the validity of such a description. Indeed, the theoretical predictions of photon condensation have been hindered by the use of truncated models lacking gauge invariance, leading to inaccurate results.

To tackle this issue, Refs. [27, 28] considered an underlying microscopic model without relying on any truncation and proved that photon condensation is prohibited as long as a single-mode spatially *uniform* vector potential is considered. In order to reproduce this result within a truncated model, it is crucial to use a gauge-invariant descriptions of the light-matter interaction such as the Peierls phase and its extensions [29–31].

More recent works [32–37] have relaxed the strong assumption of the spatially uniform

vector potential and show that photon condensation is analogous to the Condon magnetostatic instability [38]. According to these studies, photon condensation can occur only in presence of a magnetic field, while a purely electric field cannot condense as a results of no-go theorems [27, 28]. As a corollary, photon condensation is prohibited in a strictly one dimensional (1D) geometry [27,39] where the orbital motion of electrons cannot be affected by a magnetic field. Therefore, one needs to consider at least two dimension or the spin degree of freedom [36].

Here we investigate the occurrence of photon condensation in a minimal setting beyond 1D – i.e., a two-leg ladder [40–44] – where the orbital motion of spinless fermions is coupled through Peierls substitution to a non-uniform cavity mode which generates a fluctuating uniform magnetic field. Similar cavity set-ups have been proposed in Refs. [45, 46]. Moreover, recent developments have demonstrated ultra-strong coupling between magnons and the magnetic field of a superconducting resonator [47]. In contrast to 1D chains, two-leg ladders allow us to analyze transverse response to non-uniform vector potentials, while still being amenable to a thorough numerical investigation beyond typical mean-field approximations by means of the density-matrix renormalization group (DMRG) techniques [48–51] (recently being also employed in cavity quantum electrodynamics (QED) systems [52–57]).

Our results show that ladder geometries can indeed host an equilibrium superradiant transition (or photon condensation [27]), not to be confused with the non-equilibrium phase transition observed in dye filled microcavities [58]), via a first-order transition from a normal metallic phase. The first order nature of this transition arises from the strongly non-linear orbital paramagnetic response of the ladder system and provides therefore a different scenario for condensation with respect to those discussed so far in the literature [33, 34, 36]. While a photon mean-field (PMF) decoupling of the photon and matter degrees of freedom captures qualitatively the phase transition, we find that for finite sizes the correct treatment of quantum fluctuations is essential to estimate physically relevant quantities, such as current and photon properties. This demonstrates how, in these settings, the light-matter entanglement and photon squeezing cannot, in general, be neglected. In the thermodynamic limit, we show that the photon condensation allows to modify the properties of an extensive system with a single cavity mode in the collective strong coupling regime. Remarkably, even in this thermodynamic limit, where the photon state is Gaussian, it is necessary to consider both the light-matter entanglement and photon squeezing to determine the photon properties.

The structure of the paper is the following. In Sec. 2 we describe the Hamiltonian of the light-matter coupled system and introduce the main physical gauge-invariant quantities. In Sec. 3 we first introduce the PMF approximation and the DMRG numerics. Then we discuss the result comparing the two approaches and recover a qualitative agreement between the two by adding quantum fluctuations on top of the PMF solution. In Sec. 4 we move to the triangular ladder geometry highlighting the similarities with the square ladder case. In Sec. 5, we draw the conclusions and discuss possible future directions.

## 2 Hamiltonian

We consider a hybrid light-matter system where the light component is represented by a single cavity mode and the matter component is described by a tight-binding model of charged ($q = -1$) spinless free fermions on a ladder geometry. The ladder sits on the $x - y$ plane, extends in the $x$ direction with a lattice spacing $d$ and the spacing between the two legs is also $d$. Depending on the alignment of the sites on the two legs of the ladder and on the nature of inter-leg hoppings, we consider either a square or triangular geometry, see Fig. 1. The



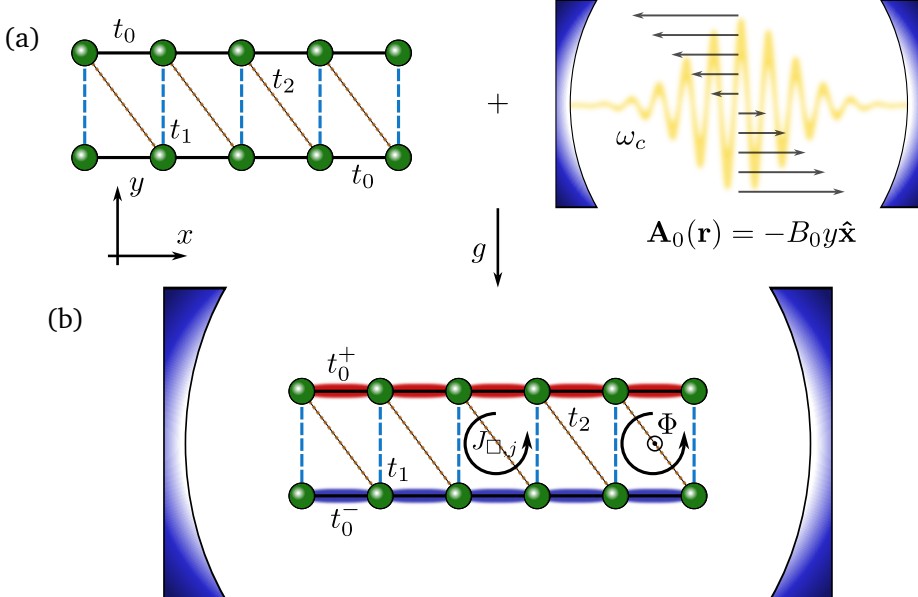

Figure 1: Sketch of the ladder plus cavity system under consideration. (a) Without any coupling to the cavity, the two legs of the ladder have the same intra-leg hopping $t_0$ (solid lines), an inter-leg hopping between corresponding site $t_1$ (dashed lines), and (for the triangular ladder) a diagonal inter-leg hopping $t_2$ (dotted lines). The cavity mode has frequency $\omega_c$ and a space profile given by the vector potential $\mathbf{A}_0(\mathbf{r}) = -B_0 y \hat{\mathbf{x}}$. (b) Upon coupling to the cavity, the intra-leg hopping terms are modified by the photon in a different way for the top ($t_0^+$) and bottom ($t_0^-$) leg.

Hamiltonian describing the fermion dynamics reads:

$$\hat{H}_0 = \left( \sum_{\sigma=\pm} \hat{H}_\sigma \right) + \hat{H}_\perp, \tag{1}$$

$$\hat{H}_\sigma = -t_0 \sum_{j=1}^{L-1} \hat{c}_{\sigma,j}^\dagger \hat{c}_{\sigma,j+1}, \tag{2}$$

$$\hat{H}_\perp = \left( -t_1 \sum_{j=1}^{L} \hat{c}_{+,j}^\dagger \hat{c}_{-,j} - t_2 \sum_{j=1}^{L-1} \hat{c}_{+,j}^\dagger \hat{c}_{-,j+1} + \text{h.c.} \right), \tag{3}$$

where $\sigma = +(-)$ indicates the top (bottom) leg of the ladder, $j = 1, \ldots, L$ is the site/rung index on each leg, and $\hat{c}_{\sigma,j}^\dagger$ ($\hat{c}_{\sigma,j}$) creates (destroys) a fermion on the site $j$ and on the leg $\sigma$. We consider open boundary conditions and one fermion per rung so that $N = L$, unless specified otherwise. Moreover, we set equal hopping amplitudes $t_0 = t_1$, while $t_2 = 0$ or $t_2 = t_0$ for the square and the triangular geometry, respectively.

The cavity setup we consider is that of a single mode where the cavity Hamiltonian is represented by a single quadratic bosonic mode with frequency $\omega_c$:

$$\hat{H}_c = \omega_c \hat{a}^\dagger \hat{a}. \tag{4}$$

Correspondingly, the cavity vector potential is $\hat{\mathbf{A}}(\mathbf{r}) = \mathbf{A}_0(\mathbf{r})(\hat{a} + \hat{a}^\dagger)$ where $\mathbf{A}_0(\mathbf{r})$ retains the spatial structure of the cavity mode and $\hat{a}$ is the annihilation operator for a photon in this cavity mode. We consider a spatially varying mode function that in the vicinity of the ladder can be written as $\mathbf{A}_0(\mathbf{r}) = -B_0 y \hat{\mathbf{x}}$. The cavity is, therefore, magnetic since the cavity mode has a non-zero curl which in classical electrodynamics gives rise to a magnetic field. In our quantum light model, this means that cavity photons generate a fluctuating magnetic flux through the

ladder plaquettes. We remark here that the single-mode approximation is not always valid and it in general depends on the specifics of the system [1,3,16–22,39,59]. In order to have a gauge-invariant coupling between matter and light, we implement the light-matter coupling by means of the Peierls substitution:

$$\hat{c}_{i,\sigma}^{\dagger}\hat{c}_{j,\sigma'} \rightarrow \exp\Big[iq\int_{R_{i,\sigma}}^{R_{j,\sigma'}} d\mathbf{r}\cdot\hat{\mathbf{A}}(\mathbf{r})\Big]\hat{c}_{i,\sigma}^{\dagger}\hat{c}_{j,\sigma'}\,, \tag{5}$$

where $R_{i,\sigma}$ denotes the position of the electronic site $i,\sigma$. In our case, the Peierls phase is non-zero only for intra-leg hoppings which are along the $x$ direction. The Peierls phase as discussed by Luttinger [60] is only an approximation of the coupling to electromagnetic fields when the value of the magnetic flux over an area, comparable to the typical size of the fermionic orbitals, is comparable to $\pi$ [29]. However the corrections strongly depend on the nature of the localized orbitals, and since neglecting these corrections does not spoil the gauge-invariant properties of the coupling, we keep only the Peierls phase.

The full light-matter coupled Hamiltonian then reads:

$$\hat{H} = \omega_{\mathrm{c}}\hat{a}^{\dagger}\hat{a}-\Big(t_1\sum_{j=1}^{L}\hat{c}_{+,j}^{\dagger}\hat{c}_{-,j} + t_2\sum_{j=1}^{L-1}\hat{c}_{+,j}^{\dagger}\hat{c}_{-,j+1}$$
$$+ t_0\sum_{j=1}^{L-1}\sum_{\sigma=\pm}e^{i\sigma g(\hat{a}+\hat{a}^{\dagger})/\sqrt{L}}\hat{c}_{\sigma,j}^{\dagger}\hat{c}_{\sigma,j+1} + \mathrm{h.c.}\Big)\,, \tag{6}$$

where we have introduced the dimensionless coupling constant $g = |q|d^2 B_0\sqrt{L}/2$ which is the parameter that drives the transition. Note that $g$ does not grow explicitly with $L$ given the scaling of the field intensity $B_0 \propto 1/\sqrt{L}$ provided that the density $N/V = L/V$ is fixed. In optical cavities, the frequency of the mode $\omega_{\mathrm{c}}$ and the field intensity $B_0$ are, in general, not independent parameters ($\omega_{\mathrm{c}} \propto B_0^2$). Still we can, in principle, tune the light-matter interaction strength $g$ independently of $\omega_{\mathrm{c}}$, for example, by varying the fermionic charge $q$. In the following we will in any case stick with $q = -1$ and use $g$ as an independent parameter[1]. The Hamiltonian (6) is invariant under the combined application of (1) the parity transformation of the photon $P_{\mathrm{ph}} : \hat{a} \rightarrow -\hat{a}$ and (2) the leg inversion $P_{\sigma} : \sigma \rightarrow -\sigma$, so that (c.f. [44])[2]:

$$\mathcal{P} \equiv P_{\mathrm{ph}}P_{\sigma}\,, \quad \mathcal{P}\hat{H}\mathcal{P}^{-1} = \hat{H}\,. \tag{7}$$

We now define two important quantities that are physically related in this light-matter system. The first one is the magnetic flux per plaquette $\hat{\Phi}$ pointing in the $z$ direction:

$$\hat{\Phi} = \int_{\square} dx dy\,\nabla\times\hat{A}(r) = \frac{2g}{\sqrt{L}}(\hat{a}+\hat{a}^{\dagger})\,, \tag{8}$$

where the $\square$ indicates the integral on a plaquette. The light-matter coupling in the Hamiltonian (6) only depends on the magnetic flux $\hat{\Phi}$ which is a well-defined physical (and thus gauge-invariant) quantity. The second quantity is the chiral charge current:

$$\hat{J}_{\chi} = \sum_{j=1}^{L-1}\hat{J}_{\square,j}\,. \tag{9}$$

---

[1]One could think of changing the lattice spacing $d$, but this would in turn change the hopping integrals. This is one of the main issue of the Peierls phase, it inevitably links the light-matter interaction and hopping integrals as discussed in [31].

[2]However, it is to be noted that independent applications of $P_{\mathrm{ph}}$ or $P_{\sigma}$ do not leave the Hamiltonian invariant.

The chiral current is defined as the sum of the plaquette currents $\hat{J}_{\square,j} = -\hat{J}_{+,j} - \hat{J}_{\perp,j+1} + \hat{J}_{-,j} + \hat{J}_{\perp,j}$ flowing in a anticlockwise direction, where $\hat{J}_{\perp,j}$ is the inter-leg current flowing from the top $\sigma = +$ to the bottom leg $\sigma = -$ at sites $j$ and $\hat{J}_{\pm,j}$ is the intra-leg longitudinal current flowing from site $j$ to site $j+1$. The gauge-invariant currents can be derived starting from the charge density $\hat{n}_{\sigma,j} \equiv q\hat{c}_{\sigma,j}^\dagger \hat{c}_{\sigma,j}$ with $q = -1$, which fulfills a discrete continuity equation $\partial_t \hat{n}_{\sigma,j} = -\hat{J}_{\sigma,j} + \hat{J}_{\sigma,j-1} - \sigma \hat{J}_{\perp,j}$. By comparing this expression with the Heisenberg equation for the density $\partial_t \hat{n}_{\sigma,j} = i[\hat{H}, \hat{n}_{\sigma,j}]$ and carrying out the explicit calculation, we find for the currents:

$$\hat{J}_{\sigma,j} = -it_0 \left( e^{i\sigma g(\hat{a} + \hat{a}^\dagger)/\sqrt{L}} \hat{c}_{\sigma,j}^\dagger \hat{c}_{\sigma,j+1} - \text{h.c.} \right), \qquad \hat{J}_{\perp,j} = -it_1 (\hat{c}_{+,j}^\dagger \hat{c}_{-,j} - \text{h.c.}).$$

Performing the sum, the contributions from the inter-leg current cancel out except for the boundary contributions and we are left with

$$\hat{J}_\chi = it_0 \sum_{j,\sigma} \left( \sigma e^{i\sigma \frac{g}{\sqrt{L}}(a + a^\dagger)} \hat{c}_{\sigma,j}^\dagger \hat{c}_{\sigma,j+1} - \text{h.c.} \right) - it_1 \left( \hat{c}_{+,1}^\dagger \hat{c}_{-,1} - \hat{c}_{+,L}^\dagger \hat{c}_{-,L} - \text{h.c.} \right). \qquad (10)$$

The chiral current and the magnetic flux operators defined above correspond to physical, gauge invariant, observables, and as such their expectation values do not depend on the choice of the gauge [61–63]. Different gauge choices are indeed implemented through unitary transformations which act on both operators and states, and leave invariant physical observables. In the following, we will use the expectation values of the chiral current and the magnetic flux as the order parameters for the photon condensation, making it a gauge-invariant phenomenon.

The presented Hamiltonian, although a minimal toy model, serves as a powerful tool in understanding the physics of magnetic photon condensation and the collective strong coupling regime of itinerant electrons coupled to a single quantized cavity mode. Despite its simplicity, realizing such a model in solid state materials embedded in optical cavities could be challenging. Nonetheless, recent advancements have demonstrated ultra-strong magnetic coupling with a superconducting resonator [47], suggesting a potential practical feasibility of our model. Another potential platform lies in cold-atom setups. However, atoms are neutral and our Hamiltonian cannot be straightforwardly realizes if not with dynamical synthetic gauge fields[3]. Here, our primary objective is to have a clearer interpretation of the results and a better understanding of the underlying physics. Therefore, we leave the question of experimental realizations open for future studies and focus on the theoretical aspects of the model in the present work.

## 3 Square ladder

We start by looking at the half-filled ($N/L = 1$) square ladder geometry ($t_2 = 0$). We solve for the ground state of the model with two approaches: *(i)* using an approximate photon mean-field decoupling where the fermionic problem is non-interacting and the light-matter entanglement is neglected; *(ii)* performing numerical simulations with DMRG where the light-matter entanglement is taken into account and the problem is fully many-body.

---

[3]Note that "dynamical" semi-classical gauge fields for cold atomic set-ups have been studied (see e.g. Ref. [64, 65]) but the dynamics is linked to their driven-dissipative nature and hence differs from the model object of this work.

## 3.1 Methods

### 3.1.1 Photon mean-field

In the photon mean-field approximation (PMF) the quantum correlations between photon and matter are neglected by assuming a product ground state $\left|\Psi^{\mathrm{PMF}}\right\rangle = |\psi_{\mathrm{m}}\rangle \left|\psi_{\mathrm{ph}}\right\rangle$ [33, 34, 59]. This gives rise to two mean-field Hamiltonians for photon $\hat{H}_{\mathrm{ph}}^{\mathrm{PMF}} \equiv \langle\psi_{\mathrm{m}}|\hat{H}|\psi_{\mathrm{m}}\rangle$ and matter $\hat{H}_{\mathrm{m}}^{\mathrm{PMF}} \equiv \left\langle\psi_{\mathrm{ph}}\middle|\hat{H}\middle|\psi_{\mathrm{ph}}\right\rangle$ that must be solved self-consistently. Up to irrelevant constants they read:

$$\hat{H}_{\mathrm{m}}^{\mathrm{PMF}} = -t_0 R \sum_{j,\sigma} e^{i\sigma\phi/2}\hat{c}_{\sigma,j}^{\dagger}\hat{c}_{\sigma,j+1} - t_1 \sum_{j}\hat{c}_{+,j}^{\dagger}\hat{c}_{-,j} + \mathrm{h.c.},\tag{11}$$

$$\hat{H}_{\mathrm{ph}}^{\mathrm{PMF}} = \omega_{\mathrm{c}}\hat{a}^{\dagger}\hat{a} + J_1 \cos\left[\frac{g(\hat{a}+\hat{a}^{\dagger})}{\sqrt{L}}\right] + J_2 \sin\left[\frac{g(\hat{a}+\hat{a}^{\dagger})}{\sqrt{L}}\right],\tag{12}$$

where we introduced the mean-field parameters $R$, $\phi$, $J_1$ and $J_2$. The first two depend on the photon state and are defined as

$$R \equiv \left|\left\langle\psi_{\mathrm{ph}}\middle|e^{i\frac{g(\hat{a}+\hat{a}^{\dagger})}{\sqrt{L}}}\middle|\psi_{\mathrm{ph}}\right\rangle\right|, \qquad \phi \equiv 2\arg\left[\left\langle\psi_{\mathrm{ph}}\middle|e^{i\frac{g(\hat{a}+\hat{a}^{\dagger})}{\sqrt{L}}}\middle|\psi_{\mathrm{ph}}\right\rangle\right],\tag{13}$$

such that $R\, e^{i\phi/2} = \left\langle\psi_{\mathrm{ph}}\middle|e^{i\frac{g(\hat{a}+\hat{a}^{\dagger})}{\sqrt{L}}}\middle|\psi_{\mathrm{ph}}\right\rangle$. The matter mean-field parameters $J_1$ and $J_2$ are defined as

$$J_1 \equiv -t_0 \sum_{j=1}^{L-1} \sum_{\sigma} \langle\psi_{\mathrm{m}}|\left(\hat{c}_{\sigma,j}^{\dagger}\hat{c}_{\sigma,j+1} + \mathrm{h.c.}\right)|\psi_{\mathrm{m}}\rangle,\tag{14}$$

$$J_2 \equiv -i t_0 \sum_{j=1}^{L-1} \sum_{\sigma} \langle\psi_{\mathrm{m}}|\left(\sigma\hat{c}_{\sigma,j}^{\dagger}\hat{c}_{\sigma,j+1} - \mathrm{h.c.}\right)|\psi_{\mathrm{m}}\rangle.\tag{15}$$

The photon parameters $\phi$ and $R$ have respectively the interpretation of a magnetic flux per plaquette and of the cavity renormalization of the hopping process. Whenever the photon quantum state is Gaussian, the expectation values of an exponential can be expressed in terms of expectation values of the two quadratures $\hat{X} \equiv (\hat{a}+\hat{a}^{\dagger})/\sqrt{2}$, $\hat{P} \equiv i(\hat{a}-\hat{a}^{\dagger})/\sqrt{2}$, and their fluctuations. In particular for Gaussian states $|\psi_{\mathrm{ph}}^{G}\rangle$, our mean-field parameters $R$ and $\phi$ become:

$$R_G = \exp\left[-\frac{2g^2}{L}\left(\langle\psi_{\mathrm{ph}}^{G}|\hat{X}^2|\psi_{\mathrm{ph}}^{G}\rangle - \langle\psi_{\mathrm{ph}}^{G}|\hat{X}|\psi_{\mathrm{ph}}^{G}\rangle^2\right)\right],\tag{16}$$

$$\phi_G = \frac{g\sqrt{2}}{\sqrt{L}}\langle\psi_{\mathrm{ph}}^{G}|\hat{X}|\psi_{\mathrm{ph}}^{G}\rangle = \left\langle\psi_{\mathrm{ph}}^{G}\middle|\hat{\Phi}\middle|\psi_{\mathrm{ph}}^{G}\right\rangle.\tag{17}$$

Note that if the photonic state is not Gaussian, in principle, we can have $\phi \neq \left\langle\psi_{\mathrm{ph}}\middle|\hat{\Phi}\middle|\psi_{\mathrm{ph}}\right\rangle$. The physical interpretation of the matter parameters $J_1$ and $J_2$ in terms of the chiral current $\hat{J}_{\chi}$ depends on the values of $\phi$ and $R$ as

$$\left\langle\Psi^{\mathrm{PMF}}\middle|\hat{J}_{\chi}\middle|\Psi^{\mathrm{PMF}}\right\rangle = R\left[J_2 \cos(\phi/2) - J_1 \sin(\phi/2)\right] + \left\langle\Psi^{\mathrm{PMF}}\middle|\hat{J}_{\perp,N} - \hat{J}_{\perp,1}\middle|\Psi^{\mathrm{PMF}}\right\rangle.\tag{18}$$

The solution of the PMF Hamiltonians is obtained by a standard self-consistent procedure:

1. Start from a guess $R$ and $\phi$;

2. Solve the single-particle problem given by the matter mean-field Hamiltonian in Eq. (11) and compute $J_1$ and $J_2$ as Eqs. (14) and (15);

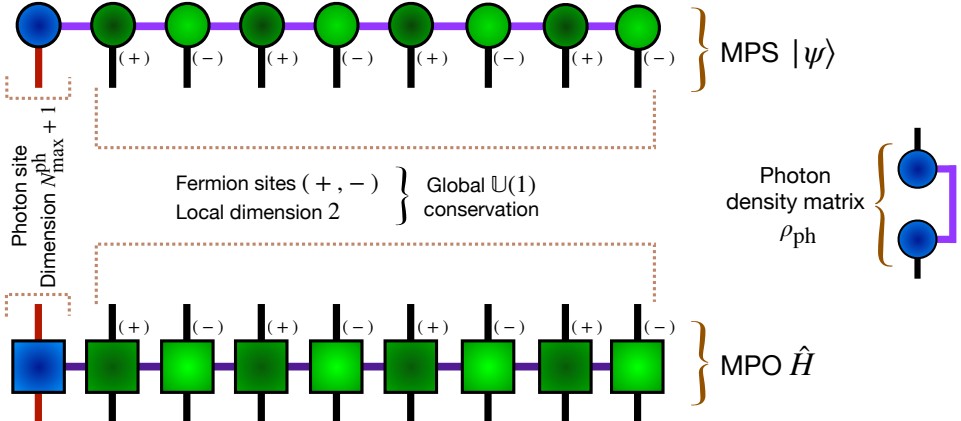

Figure 2: The matrix-product state (MPS) and the matrix-product operator (MPO) representations used in our study. The photonic Hilbert space is truncated to accommodate maximum number of photons $N_{\max}^{\mathrm{ph}} = 63$, while the fermionic local Hilbert space dimension is 2. Note that the photon mean-field (PMF) approximation is implicitly setting $\chi = 1$ for the first link that connects the photon and fermionic degrees of freedom. In this MPS representation, the photon density matrix can be computed efficiently by tracing out the fermionic degrees of freedom.

3. Solve the photonic mean-field Hamiltonian in Eq. (12) via exact diagonalization and compute a new $R'$ and $\phi'$ from Eq. (13);

4. Repeat from 2, using $R'$ and $\phi'$ as a new mean-field parameters until the desired convergence is reached.

Note that in presence of first-order transitions one needs to be careful and try different initial guesses as the self-consistency can get stuck in local minima of the energy.

### 3.1.2 Density-matrix renormalization group

This ladder geometry, being a quasi-1D system, is well-suited to be approached via the density-matrix renormalization (DMRG) techniques [48–51]. The matrix-product state (MPS) representation that we use for this purpose is similar to those employed in previous works on light-matter systems [52–55], where the single photon site is placed at one end of the MPS chain, while rest of the MPS is made up of fermionic sites, as depicted in Fig. 2. Additionally, to preserve the global $\mathbb{U}(1)$ symmetry associated with the conservation of total fermionic charge $\sum_{\sigma,j} n_{\sigma,j}$, we employ $\mathbb{U}(1)$ symmetric tensors [66,67] for the fermionic sites, while a standard dense tensor is used for the photon site. Moreover, the use of the matrix-product operator (MPO) representation for the Hamiltonian, as illustrated in Fig. 2, is efficient as the long-range light-matter interaction term can be expressed exactly in the MPO form [68,69]. The ground state obtained through DMRG is a variationally computed state, with an error that can be precisely controlled through the bond dimension $\chi$ of the MPS ansatz. By adjusting the bond dimension, one can verify the convergence and attain the desired level of accuracy. See App. D for further details on the convergence of DMRG simulations. For the numerical implementation of the DMRG algorithm we use the ITensor library [70] and the respective codes can be found at GitHub[4].

---

[4]The example codes for this work can be found at the GitHub repository: https://github.com/zenobacciconi/cavity_ladder.git.

It is important to note that, in a symmetry-broken phase, while strictly speaking, exact symmetry breaking does not occur in the ground state at finite sizes, it is a well-established characteristic of DMRG to converge to one of the symmetry-broken states, as these states have significantly less entanglement compared to the macroscopic superposition of two (or more) symmetry-broken states. In the following discussion, while considering the symmetry-broken phase, we will focus solely on these symmetry-broken ground states, which can be reached either automatically through the DMRG algorithm or with the aid of a small symmetry breaking term[5].

The DMRG solution allows us then to easily access the photon density matrix by tracing out the matter degree of freedom as (see Fig. 2):

$$\rho_{\text{ph}} = \text{Tr}_{\text{m}}\Big[ |\Psi\rangle \langle\Psi| \Big]. \tag{19}$$

From this we can, for example, calculate the entanglement entropy of the cavity with respect to matter $S(\rho_{\text{ph}}) = -\text{Tr}\Big[\rho_{\text{ph}} \ln \rho_{\text{ph}}\Big]$ that in the PMF decoupling is exactly zero.

## 3.2 Results: Photon mean-field vs. DMRG

**Phase diagram.** A sample of the results obtained with DMRG and PMF are depicted in Fig. 3. Overall, we observe that the two approaches match in determining the structure of the phase diagram – both show a first-order phase transition from a normal metallic phase for $g < g_{\text{c}}$ to a photon condensed phase for $g > g_{\text{c}}$ characterized by a $\mathbb{Z}_2$ symmetry breaking where the symmetry $\mathcal{P}$ (see Eq. (7)) gets spontaneously broken. The energy kink shown in Fig. 3(d) support the first-order nature of the transition for both DMRG[6] and PMF. The normal phase is connected to the state at $g = 0$ and display metallic properties. The photon condensed phase is a Condon phase where a finite current $\langle \hat{J}_\chi \rangle \neq 0$ (Fig. 3(b)) is linked to a finite magnetic flux $\langle \hat{\Phi} \rangle \neq 0$ (Fig. 3(a))[7]. In particular, we have $|\langle \hat{J}_\chi \rangle| = \frac{L}{2}(\pi - |\langle \hat{\Phi} \rangle|)$ from both DMRG and PMF with small finite-size corrections, where $|\langle \hat{\Phi} \rangle| \leq \pi$ and $\langle \hat{J}_\chi \rangle$ and $\langle \hat{\Phi} \rangle$ have the same sign. The matter state is a diamagnetic band insulator, also called the Hofstadter flux state in the context of fermionic ladder with static magnetic fields [41]. The notion of diamagnetism, however, in the present context is an unusual one. While usually one defines diamagnetism when the magnetization of a material is opposite with respect to an applied magnetic field, here the magnetization is in the same direction but proportional to the difference $\pi - |\langle \hat{\Phi} \rangle|$. Diamagnetism must be interpreted as a response of the system trying to bring the magnetic flux not to 0 but to $\pi$.

**Finite size effects.** The chiral current shown in Fig. 3(b) has finite size corrections which compare well between PMF and DMRG, but the exact transition point is shifted towards lower

---

[5]For large enough system-size $L$, DMRG may randomly converge to one of the symmetry-broken ground states, which can also be influenced by the choice of initial input state. To eliminate such arbitrariness, we add a very small symmetry breaking term in our simulations and select only a specific symmetry-broken state.

[6]To mitigate the negative impacts of metastability near the first-order phase transition, we employ a two-fold approach for each value of $g$ within a range close to $g_c$. We simultaneously run two separate DMRG simulations, each initialized with a state deep within a distinct phase. From these two simulations, we select the outcome that results in the lowest energy.

[7]It is straightforward to check that under the symmetry operation $\mathcal{P}$, $\langle \hat{J}\chi \rangle$ and $\langle \hat{\Phi} \rangle$ change their sign. Therefore, all the symmetry-preserving eigenstates must have vanishing chiral current and magnetic flux. In the photon condensed phase, the $\mathbb{Z}_2$ symmetry associated with $\mathcal{P}$ gets spontaneously broken, and the ground state exhibits a two-fold degeneracy. This degeneracy arises from the presence of two symmetry-broken states, denoted by $|\Psi_+\rangle$ and $|\Psi_-\rangle$, which have positive and negative values of the order parameters $\langle \hat{J}\chi \rangle$ and $\langle \hat{\Phi} \rangle$ respectively. Under the symmetry operation, we have $\mathcal{P}|\Psi_\pm\rangle = |\Psi_\mp\rangle$, so that the symmetric and anti-symmetric combinations $\frac{1}{\sqrt{2}}(|\Psi_+\rangle \pm |\Psi_-\rangle)$ belong to the even and odd symmetry sectors respectively, each having zero magnetic flux and chiral current.

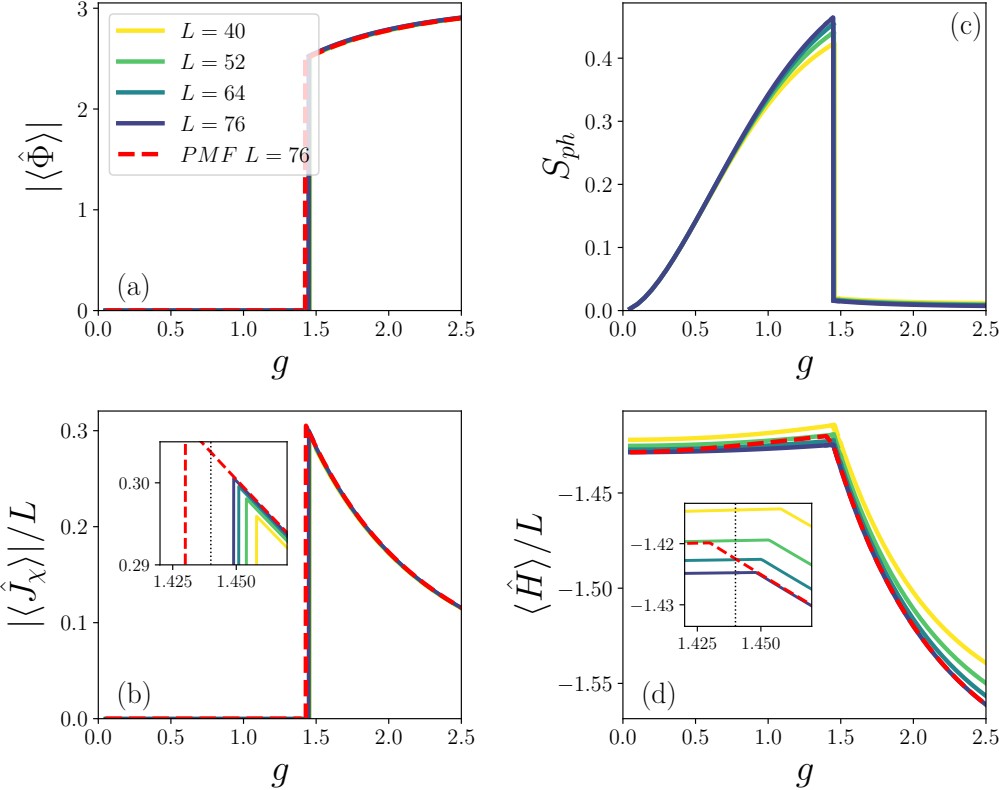

Figure 3: DMRG results for different system sizes and their comparison with the photon mean-field (red dashed line). (a-b) The magnetic flux per plaquette and chiral current as the cavity and the matter order parameters respectively. They both show a discontinuity at the first-order transition point $g_c \simeq 1.44$ that slightly shifts at finite sizes. The relation $|\langle \hat{J}_\chi \rangle| = \frac{L}{2}(\pi - |\langle \hat{\Phi} \rangle|)$ is satisfied in the thermodynamic limit. Note that here we plot the absolute values of the order parameters since they can take on both negative and positive values in the photon condensed phase depending on the degenerate symmetry-broken state, but always have the same sign as each other. (Inset) Zoom near the thermodynamic limit transition point $g_c = 1.44$ marked with a vertical black line. Note that finite size corrections to the transition point and value of the current are different between DMRG and PMF. (c) Photon entanglement entropy with respect to the matter. It is finite in the thermodynamic limit for both phases, although much smaller in the photon condensed phase. (d) Total energy density showing a kink at the transition point as expected for a first-order transition both in DMRG and the photon mean-field. Note that finite size corrections are different.

(PMF) and higher (DMRG) values of the coupling strength. The reason is that while the photon condensed phase has low photon entanglement (Fig. 3) and is well captured by the PMF, the normal phase has high photon entanglement and then finite-size corrections are different between PMF and DMRG. In particular the finite size effect of the PMF comes mainly from the mean-field hopping renormalization $R$ that tends to 1 in the thermodynamic limit as the squeezing of the cavity remain finite (Eq. (16)). The same happens for the total energy and the magnetic susceptibility (not shown).

**Magnetostatic instability.** We refer to the instability to a ground-state displaying a finite magnetic flux $\langle \hat{\Phi} \rangle \neq 0$ as a "magnetostatic instability" [32, 33, 37, 71]. In Fig. 4 we show the mean-field picture of the magnetostatic instability characterizing the first-order transition.

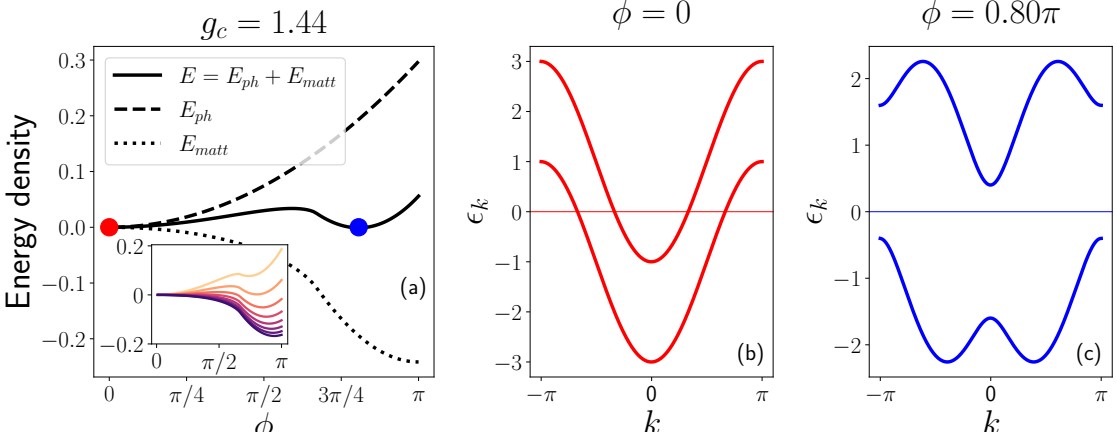

Figure 4: The mean-field picture of the transition. (a) Mean-field energy per particle as a function of a cavity generated 'classical' magnetic flux $\phi$ at $g = g_c = 1.44$. The photon energy is $E_{\text{ph}} = \omega_c(\phi/4g)^2$ and the matter energy $E_{\text{matt}} = \langle \psi_{\text{m}} | \hat{H}_{\text{m}}^{PMF} | \psi_{\text{m}} \rangle / L$ is obtained from the single-particle problem of Eq. (11) fixing $R = 1$. (Inset) $E(\phi)$ shown from $g = 1.2$ (light color) to $g = 2.8$ (dark color) across the transition. After the transition the solution at $\phi = 0$ does not immediately become unstable, signaling metastability – a typical feature of first-order phase transitions. (b) Metallic and (c) insulating band structure at two fluxes corresponding to the two minima of the total energy at $g = g_c$ in periodic boundary conditions. Horizontal lines mark the chemical potential.

Differently to a second-order transition [33] the instability is not controlled by the linear orbital magnetic susceptibility (which is a property of the Fermi-surface [34]) of the normal state $\phi = 0$. Conversely, the first-order nature is given by a strong non-linear response of the system at strong magnetic fluxes. In particular, the non-linear behavior comes together with an indirect gap opening in the band structure at $\phi = 2\pi/3$. Once fixed the fermionic bare energies, the transition point only depends on the energy of the cavity per unit of flux, which is controlled by the combination $\omega_c/g^2$ as the cavity PMF energy density in the thermodynamic limit is $E_{\text{ph}} = \omega_c(\phi/4g)^2$.

The numerical results from DMRG simulations confirm the photon mean-field picture for the instability. Our findings show that the only way for a single cavity mode in the collective strong coupling regime to change the macroscopic properties of a thermodynamically large system is via a macroscopic classical state. Quantum fluctuations and light-matter entanglement give only $O(1)$ contributions [59, 72, 73], which can be important near quantum criticality, as recently discussed in Ref. [74].

**Cavity quantum state.** As shown in Fig. 5, the density matrix $\rho_{\text{ph}}$ obtained via DMRG shows that the cavity can be can be accurately approximated by a Gaussian state (see App. A). In order to quantify the non-Gaussianity of the state, we compute the Quantum relative entropy [75]:

$$\Delta_G = S(\rho_{\text{ph}}^{\text{G}}) - S(\rho_{\text{ph}}) \,, \tag{20}$$

where $\rho_{\text{ph}}^{\text{G}}$ is the Gaussian density matrix that has the same expectation values $\langle \hat{X}^2 \rangle$, $\langle \hat{P}^2 \rangle$, $\langle \hat{X} \rangle$, $\langle \hat{P} \rangle$ of $\rho_{\text{ph}}$. The non-Gaussian nature of the state is a finite size correction (see Fig. 5(a)) and it arises from the non-linear nature of the Peierls phase. The non-Gaussianity revealed to be sensitive to numerical details at sizes higher than $L = 76$ for which a more careful analysis is needed (App. D). Then in order to have more information on the nature of the corrections we

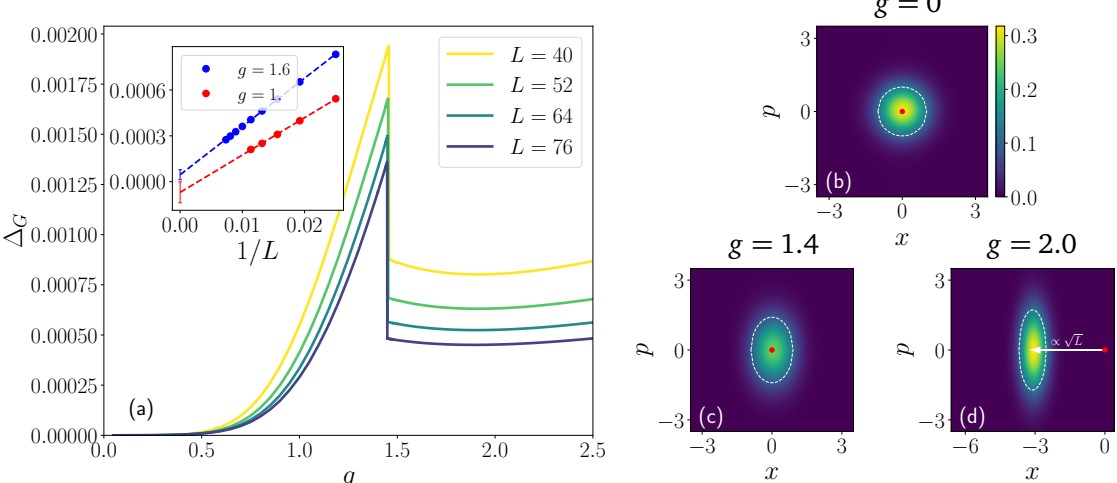

Figure 5: Cavity quantum state obtained with DMRG. (a) Quantum relative entropy $\Delta_G$ as a measure of cavity non-Gaussianity for different system sizes. The state is Gaussian in the thermodynamic limit in both phases. (Inset) The numerical extrapolation of the non-Gaussianity to the thermodynamic limit by using a linear fit $f(1/L) = a + b/L$. Errors of $10^{-5}$ on the DMRG points have been considered to account for slight dependency on numerical parameters (see App. D). (b-d) Wigner functions of the photon at finite system-size $L = 40$ for the uncoupled system $g = 0$ (b), in normal phase $g = 1.4$ (c), and in the photon condensed phase $g = 2$ (d). Even at the smallest presented size when the non-Gaussianity is larger, the Wigner function is positive everywhere. White contours display the width of the corresponding Gaussian state $\rho_{\text{ph}}^G$ having same $\langle \hat{X}^2 \rangle - \langle \hat{X} \rangle^2$ and $\langle \hat{P}^2 \rangle - \langle \hat{P} \rangle^2$ as the actual photon state $\rho_{\text{ph}}$, and the white arrow shows the displaced nature of the photon condensed state. Red dots mark the origin $(x, p) = (0, 0)$.

also show, in Figs. 5(b)-(d), the Wigner function [76] of the cavity at the smallest investigated system size. The corrections do not spoil the positivity of the Wigner function nor the qualitative shape. Also in the PMF (not shown) the non-Gaussianity goes to zero in the $L \to \infty$ and the squeezing remain constant so that $R \to 1$. The dashed lines in Figs. 5(b)-(d) mark the width of the respective Gaussian states and they encode the nature of the Gaussian state. A finite light-matter entanglement is represented by a lager area enclosed in the dashed lines as it increase the variance of both quadratures (see App. A), while squeezing reduce the fluctuations in one quadrature while increasing the other one to keep the product constant. The gaussianity of the cavity state can be an important starting point for semiclassical treatments of light-matter problems [77]

## 3.3 Results: Gaussian fluctuations

In order to gain further insights into the entangled light-matter ground state one can try to calculate pertubative contribution to the ground state $|\Psi_0\rangle$ in powers of $g$. Leaving a more detailed discussion in App. C at first order (with periodic boundary conditions) we obtain:

$$\left| \Psi_{(1)} \right\rangle = \left| \Psi_0 \right\rangle - \frac{g}{\sqrt{L}} \sum_k \frac{2 f_k}{\omega_k + \omega_c} \left| \text{PH}_k, 1_c \right\rangle , \tag{21}$$

where $|\text{PH}_k, 1_c\rangle = \hat{\tilde{c}}_{k,2}^\dagger \hat{\tilde{c}}_{k,1} \hat{a}^\dagger |\Psi_0\rangle$ is a polaritonic state with one photon in the cavity and with one direct particle-hole excitation over the Fermi see at momentum $k$, $\hat{\tilde{c}}_{k,a}$ with $a = 1, 2$ are the fermionic operators that diagonalize $H_{\text{m}}^{\text{PMF}}$, $\omega_k$ is the direct band gap, and $f_k$ are the matrix

elements for the magnetic transition. This matches the DMRG solution at small values of $g$ (not shown), but it becomes useless at higher values of $g$, in particular after the photon condensation transition.

As an alternative method to gain analytical insight on the system dynamics, we examine the Gaussian fluctuations above the mean-field solution which, in the normal phase, at first order in $g$ actually recover the above perturbative result. This treatment is motivated by the observation that, for all values of $g$ considered, the photon state is always Gaussian in the thermodynamic limit. We then expand the photon operator as:

$$\hat{a} = \alpha_0 + \delta\hat{a}, \tag{22}$$

where $\alpha_0 = \phi\sqrt{L}/4g$ is the solution of the photon mean-field decoupling restricted to coherent states and $\delta\hat{a}$ the bosonic quantum fluctuations around it fulfilling the bosonic commutation relations $[\delta\hat{a}, \delta\hat{a}^\dagger] = 1$. Note that the coherent state around which we are expanding does not, in general, correspond to the full solution of the PMF as that is a generic pure quantum state of the cavity Hilbert space.

To simplify the treatment, we work in periodic boundary conditions (details in App. B) and introduce the creation operator in momentum space $\hat{c}_{\sigma,k} = \frac{1}{\sqrt{L}}\sum_j e^{-ikj}\hat{c}_{\sigma,j}$. and the pseudospin representation $\hat{\sigma}_k^\alpha = (\hat{c}_{+,k}^\dagger \quad \hat{c}_{-,k}^\dagger)\sigma^\alpha(\hat{c}_{+,k} \quad \hat{c}_{-,k})^T$ with $\sigma^\alpha$ being the Pauli matrices ($\alpha = 0,1,2,3$). Now in order to obtain a quadratic Hamiltonian, we expand the Peierls phase up to second order in $\delta\hat{a}/\sqrt{L}$ obtaining up to constants:

$$\hat{H} \simeq \omega_c \delta\hat{a}^\dagger\delta\hat{a} + \sum_{k,\alpha}\hat{\sigma}_k^\alpha h_k^\alpha + \frac{g}{\sqrt{L}}(\delta\hat{a} + \delta\hat{a}^\dagger)\sum_{k,\alpha}\hat{\sigma}_k^\alpha d_{p,k}^\alpha - \frac{g}{2L}(\delta\hat{a} + \delta\hat{a}^\dagger)^2\sum_{k,\alpha}\hat{\sigma}_k^\alpha d_{d,k}^\alpha, \tag{23}$$

where

$$\begin{aligned}
\boldsymbol{h}_k &= -(2t_0\cos(k)\cos(\phi/2), t_1, 0, 2t_0\sin(k)\sin(\phi/2)), \\
\boldsymbol{d}_{p,k} &= -(-2t_0\cos(k)\sin(\phi/2), 0, 0, 2t_0\sin(k)\cos(\phi/2)), \\
\boldsymbol{d}_{d,k} &= -(2t_0\cos(k)\cos(\phi/2), 0, 0, 2t_0\sin(k)\sin(\phi/2)).
\end{aligned} \tag{24}$$

Since the light-matter coupling remains diagonal in the momentum space, the occupation will be conserved at each momentum, and it will be $N_k = \langle\hat{\sigma}_k^0\rangle = 0,1,2$ depending on the mean-field solution encoded in $\phi$. The only non-trivial momentum sectors are those singly occupied where a direct particle-hole excitation is allowed, for the others $\langle\hat{\sigma}_k^{1,2,3}\rangle = 0$. For every momentum sector with occupation $N_k = 1$, we can rotate the pseudospin degree of freedom to a new basis $\hat{\tilde{\sigma}}_k$, so that every term in the Hamiltonian, except for the ones containing cavity fluctuations $\delta\hat{a}$, becomes diagonal. Then we use a Holstein-Primakoff transformation of the particle-hole pseudospin for which $\hat{\tilde{\sigma}}_k^3 = -(1-2\hat{b}_k^\dagger\hat{b}_k)$ and $\hat{\tilde{\sigma}}_k^1 = (\hat{b}_k + \hat{b}_k^\dagger)$ which is exact if one consider $\hat{b}_k$ as hard core bosons. Since the occupation of a single particle-hole is not expected to be more than $O(1/L)$ we lift the hard core boson constraint. As a last step, we discard non-quadratic terms to obtain a quadratic Hamiltonian:

$$\hat{H}^{(2)} = \omega_c\delta\hat{a}^\dagger\delta\hat{a} + \sum_k \omega_k\hat{b}_k^\dagger\hat{b}_k - D\frac{g^2}{2}\left(\delta\hat{a} + \delta\hat{a}^\dagger\right)^2 + g\left(\delta\hat{a} + \delta\hat{a}^\dagger\right)\sum_k P_k\left(\hat{b}_k + \hat{b}_k^\dagger\right), \tag{25}$$

with

$$\omega_k = 2\sqrt{\sum_{\alpha=1}^3 (h_k^\alpha)^2}, \qquad P_k = \delta_{N_k,1}\frac{2}{\sqrt{L}}\frac{d_{p,k}^3 d_{\alpha,k}^1}{\omega_k},$$

$$D = \frac{1}{L}\sum_k N_k d_{d,k}^0 - \frac{2}{L}\sum_k \delta_{N_k,1}\frac{d_{d,k}^3 d_{d,k}^3}{\omega_k}. \tag{26}$$

The quadratic Hamiltonian in Eq. (25) can be diagonalized with a Hopfield-Bogoliubov transformation [78] obtaining:

$$\hat{H}^{(2)} = E_0 + \sum_{\mu=1}^{M+1} \epsilon_\mu \hat{d}_\mu^\dagger \hat{d}_\mu, \tag{27}$$

where $\hat{d}_\mu$ is the polariton annihilation operator and

$$M = \sum_k \delta_{N_k,1} \tag{28}$$

is the number of available particle-hole transition which depends on the mean-field phase. The latter is $M = L/3$ in the normal phase and $M = L$ in photon condensed phase. The vacuum of polaritons, defined by $\hat{d}_\mu |0_{\text{pol}}\rangle = 0$ for all $\mu$, corresponds to a multi-mode Gaussian state of cavity photon and particle-hole excitations, and is different from the mean-field ground state $|\Psi^{PMF}\rangle \neq |0_{\text{pol}}\rangle$. In the limit of small $g$, the ground state wavefunction $|0_{\text{pol}}\rangle$ coincides with the first-order perturbative result of Eq. (21), and then corrects it with $O(g^2)$ terms that give rise to the squeezing of the cavity mode. In order to check the validity of this treatment, we can directly compare the photon observables in the thermodynamic limit (Fig. 6). Note that finite system comparisons should be done carefully as finite size corrections arise both from different boundary conditions and from higher order terms discarded in Eq. (25). Minor discrepancies appear to emerge in the thermodynamic limit, but these are likely due to insufficiently large sizes in our numerical results. Since the system is non-additive due to the cavity's presence, we cannot rule out non-trivial corrections that may not be captured by straightforward $1/L$ extrapolations, particularly for non-linear observables such as entanglement entropy. However, this does not contradict the result presented in Fig. 5, as it is, in general, unable to spoil the Gaussianity of the cavity density matrix. Nonetheless, up to this minimal errors, the treatment of Gaussian fluctuations is able to faithfully capture the nature of the light-matter correlated ground state (see the comparisons in Fig. 6).

Now given the quadratic Hamiltonian in Eq. (27) we can also get information on the excited states of the light-matter system. In particular, we show, in Fig. 7, the zero temperature spectral function of the photon, calculated in the Lehmann representation as:

$$A(\omega) = \sum_{\mu=0}^{M+1} \left| \langle \mu | \hat{a}^\dagger | 0_{\text{pol}} \rangle \right|^2 \delta(\omega - \epsilon_\mu) - \left| \langle \mu | \hat{a} | 0_{\text{pol}} \rangle \right|^2 \delta(\omega + \epsilon_\mu), \tag{29}$$

where $|\mu\rangle = \hat{d}_\mu^\dagger |0_{\text{pol}}\rangle$ for $\mu > 0$ and $\mu = 0$ corresponds to the vacuum $|0_{\text{pol}}\rangle$. In the normal phase $g < g_c$ we clearly see two polariton lines. The lower polariton starts at the cavity frequency $\omega_c = 1$ and the upper polariton at the energy $\omega = \omega_k = 2$ that corresponds to the excitation energy of all direct particle-hole excitations, as clear from the band structure of Fig. 4(b). The hybridized degree of freedom is then a superposition of all $M$ available particle-hole excitations leaving $M-1$ dark polariton dark states, akin to what happens for intersubband exciton-polaritons [79] where intersubband particle-hole excitations provide a macroscopic electric dipole moment. In the photon condensed phase $g > g_c$ instead one polariton mode brings almost all the photon spectral weight and crosses the rest of the polariton modes made up by the particle-hole continuum. Indeed, whether or not a clear polariton doublet can form depend on the band structure presented in figure 4(b,c). The energy of the brightest polariton in the photon condesed phase is increasing with $g$ as the cavity fluctuations are more and more squeezed due to the term proportional to $D$ in Eq. (27). Consistently with the first order nature of the transition the polariton gap does not close.

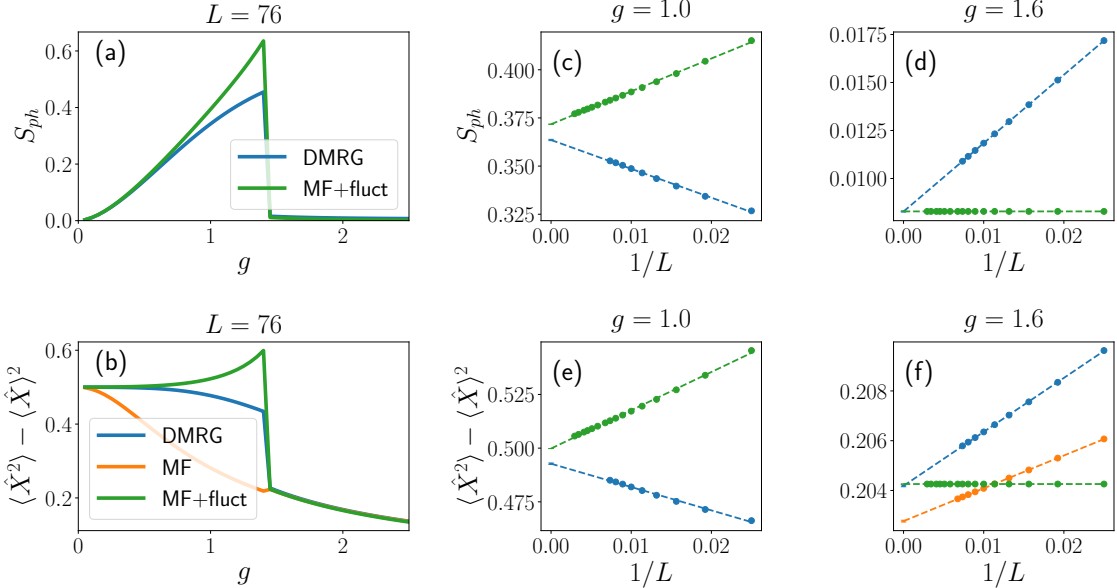

Figure 6: (a-b) The photon entropy $S_{\mathrm{ph}}$ and the variance of the quadrature $X$ for the three different methods: DMRG (red), mean-field (blue), and mean-field plus Gaussian fluctuations (green). $S_{\mathrm{ph}}$ is zero by definition for the PMF, and hence is not shown. Note that while the system-size is fixed at $L = 76$ for all three methods, the boundary condition for the treatment with Gaussian fluctuations have been set to periodic instead of open. (c-f) The scaling analysis at two representative points inside each phase: $g = 1$ (c,e) and $g = 1.6$ (d,f). Dashed lines are linear fits to a function $f(1/L) = a + b/L$, and have been extrapolated to $L \rightarrow \infty$. For $g = 1$, the extrapolation of $\langle \hat{X}^2 \rangle - \langle \hat{X} \rangle^2$ for the PMF is not plotted and its extrapolated value is $a = 0.279$. The difference with the bare mean-field is most evident in the normal phase, where the photon entanglement is higher. Discrepancies between MF+gaussian fluctuations are on the order of $10^{-3}$, hence we cannot exclude some non-trivial corrections beyond $1/L$ in particular for non-linear observables as $S_{ph}$.

## 4 Triangular ladder

We now discuss the similar case of a triangular ladder geometry with $t_0 = t_1 = t_2 = \omega_{\mathrm{c}} = 1$. We do not present explicitly all the calculations for the PMF and for the Gaussian fluctuations, as these can be done in close analogy with the square ladder case. We only mention here that the inclusion of the $t_2$ hopping changes $\boldsymbol{h}_k$, and the parameters $\omega_k$, $D$, and $P_k$ appearing in Eq. (25). As shown in Fig. 8, we find again a first order transition to a photon condensed state with $\langle \hat{\Phi} \rangle \neq 0$. The main qualitative difference is in the state of the matter which now goes from a metallic state with 4 Fermi points to a metallic state with 2 Fermi points as evident from the band structure in Fig. 8. Again the normal phase has more photon entanglement with respect to the photon condensed phase, and indeed the PMF does not correctly capture the fluctuations of the cavity quadrature $\hat{X}$ below the transition but Gaussian fluctuations are in good agreement. We remind the reader here that the treatment with Gaussian fluctuations is done with periodic boundary conditions while PMF and DMRG are with open boundary conditions. However, in the thermodynamic limit (not shown) results are compatible with the interpretation given for the square ladder case.

Now by looking at the photon spectral function it is now clear that high photon entanglement in the normal phase is not directly linked to a strong coupling to a single collective

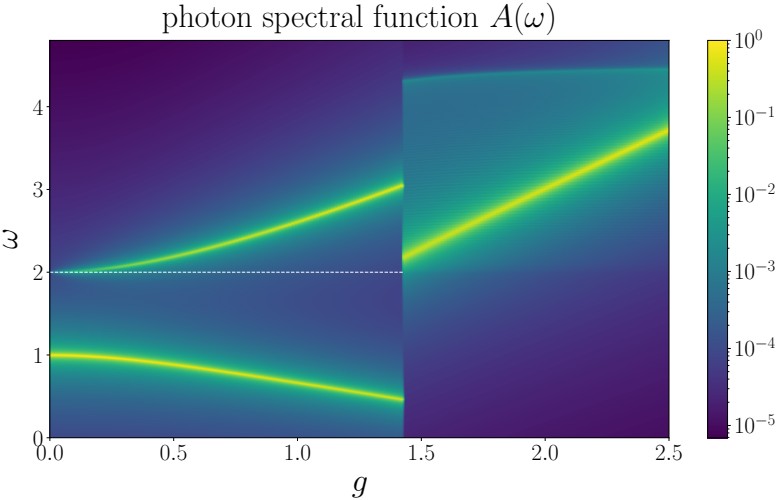

Figure 7: The photon spectral function $A(\omega)$ obtained from the treatment with Gaussian fluctuations, for the square lattice case. Each polariton energy is smeared with a Lorentzian of width $\eta = 10^{-2}$. In the normal phase, there are two bright polariton modes starting at frequencies $\omega_c = 1$ and $\omega = \omega_k = 2$, and $M - 1$ dark modes at $\omega = \omega_k = 2$ signaled by a white dashed line (see Eq. (28) for the definition of $M$). In the photon condensed phase only the one polariton mode is clearly visible while the the other one is shared between all the particle-hole excitations that now have a non-uniform energy structure. The polariton gap does not close at the first-order transition. System size here is $L = 400$.

excitation. While for the square only one collective excitation was mixing with the cavity, here it is evident that the whole continuum of particle-hole excitations is contributing as there is no polariton doublet in the spectrum at $g < g_c$. In the photon condesed phase, the ground state of the cavity is strongly squeezed and hence its excitation energy is pushed to higher frequencies as we increase $g$.

## 5 Conclusion

In this work, we have proposed a class of minimal toy models for charged fermions coupled through a Peierls phase to a non-uniform cavity mode. The cavity hosts a fluctuating magnetic flux which above a critical light-matter coupling develops a non-zero expectation value, leading to a first-order transition. The crucial element to overcome no-go theorems, as also noted in [34–36], is the presence of a magnetic coupling. To the best of our knowledge, this is the first example of an equilibrium first-order photon condensation for an electronic system. Alternative examples of first-order photon condensation have been proposed [80,81]. However, these examples do not involve itinerant electronic systems and, more importantly, have been demonstrated to be artifacts resulting from Hilbert space truncation of the model. [28]. We have shown how the key element for such transition is a strong non-linear magnetic response of the ladder band structure, coming with a sudden change of the number of Fermi points as a function of the photonic order parameter, 4 to 0 (2) for the square (triangular) ladder case. Thanks to the quasi-1D nature of the ladder geometry, we have been able to study the ground state via DMRG, hence fully taking into account light-matter entanglement and all kinds of quantum fluctuations. Our numerical results confirms that quantum fluctuations of a single cavity mode alone in the so-called collective strong coupling regime ($g = $ const for $L \to \infty$)

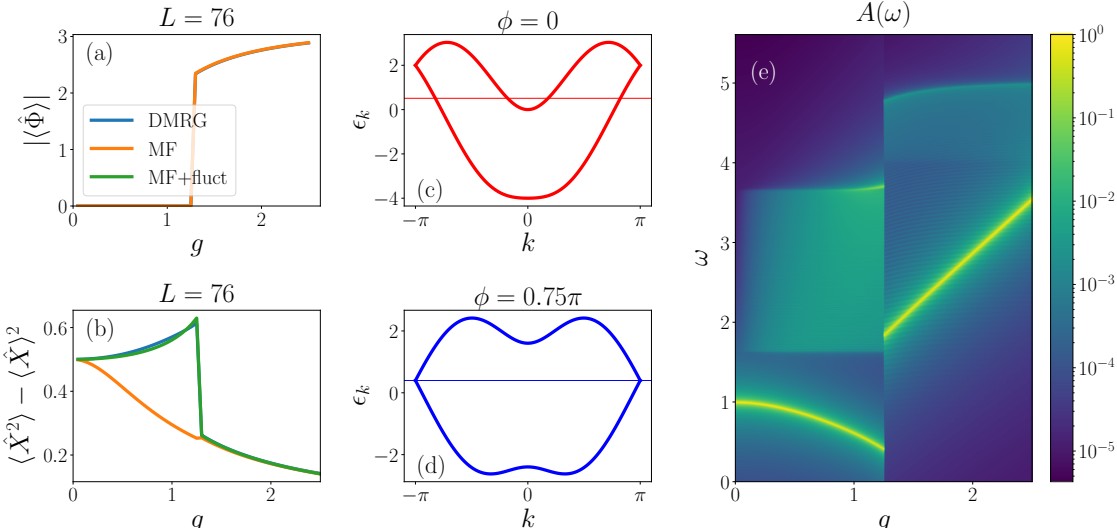

Figure 8: Results for the triangular ladder geometry. (a) The photonic order parameter showing the first-order photon condensation at $g = 1.26$ for both DMRG and PMF. (b) Variance of the cavity quadrature $\hat{X}$ with the three different methods. We show only one system size and the PMF+fluctuations is done with periodic boundary conditions while DMRG and PMF are in open boundary conditions. (c-d) Band structures of the PMF problem for fixed $R = 1$ and two values of $\phi$ corresponding to the two minima at the transition. Horizontal lines mark the chemical potential. (e) Photon spectral function obtained via the Gaussian fluctuations. Again the gap does not close at the transition due to its first order nature. Each polariton energy has been smeared with a Lorentzian of width $\eta = 10^{-2}$

do not alter the phase diagram of a thermodynamically large system [59,72] and mean-field solutions are accurate up to finite size corrections [73].

Indeed the transition we discussed is agnostic to the PMF decoupling. Still, we find that light-matter entanglement is essential to properly describe the quantum state of a strongly coupled cavity mode as discussed in Fig. 6. As already found in other systems with linear dipole-like light-matter couplings [56] the cavity state is Gaussian. Here we have shown how the non-linear nature of the Peierls phase gives a small non-Gaussian correction at finite sizes without any qualitative changes in the Wigner function that remains positive in all the explored phases.

Supported by the Gaussian nature of the cavity ground state in the thermodynamic limit, we analytically derived the quadratic fluctuations on top of the mean-field solution. This highlights the role of polariton states whose ground state gives a qualitatively correct result for the photon entropy and gives access to the cavity spectral function. The latter reflects the first-order nature of the transition.

The presented model is a valid starting point to study photon condensations for large enough system sizes in a numerically exact way. Although not shown in the main text the magnetic instability of the ladder geometry exists for a wide range of geometries and Hamiltonian parameters, including both metal-metal and insulator-insulator first and second order transitions. Interacting electrons coupled to static magnetic fields have also been investigated in higher-dimensional systems [82], showing similar first-order behaviour. This finding supports the notion that first-order photon condensation could be prevalent in various settings, and solely examining instabilities of the normal phase might be restrictive. Moreover, local fermion-fermion interactions can be included without any added cost to the DMRG simula-

tions, as recently done in [83] for a single XXZ chain. Another element that would be interesting to add to the model is the electronic spin which should favor the paramagnetic response of the system and could have a non-trivial interplay with the orbital magnetism subject of this work.

Finally, we note that a recent study [84], that appeared on the same day on arXiv, have obtained similar results in a system of Van Vleck paramagnetic molecules, showing the importance of cavities with a significant magnetic component.

# Acknowledgements

We express our sincere gratitude to G. Arwas, B. Beradze, M. Capone, I. Carusotto, C. Ciuti, D. De Bernardis, O. Di Stefano, D. Fausti, G. Mazza, A. Mercurio, C. Mora, A. Nersesyan, F.M.D Pellegrino, M. Polini and S. Savasta. for useful discussions. We are particularly grateful to the organizers of *Shedding Quantum Light on Strongly Correlated Materials (QLCM22)* where this collaboration took its first steps. Z. B. and M. D. acknowledge PNRR MUR project PE0000023-NQSTI for financial support.

**Funding information**    The work of G. C. and M. D. was partly supported by the ERC under grant number 758329 (AGEnTh), and by the MIUR Programme FARE (MEPH). T. C. acknowledges the support of PL-Grid Infrastructure for providing high-performance computing facility for a part of the numerical simulations reported here. G.M.A. and M.S. acknwoledge funding from the European Research Council (ERC) under the European Union's Horizon 2020 research and innovation programme (Grant agreement No. 101002955 – CONQUER).

# A    Gaussian states and Wigner function

Given a bosonic degree of freedom $\hat{a}$, a Gaussian state can be identified by two complex parameters $\alpha$, $\xi$, and one real positive parameter $N_{\text{th}}$. For a single mode, the density matrix of a generic Gaussian state can be written as a displaced and squeezed thermal state:

$$\rho^G = \hat{D}(\alpha)\hat{S}(\xi)\frac{N_{\text{th}}^{\hat{a}^\dagger\hat{a}}}{(1+N_{\text{th}})^{\hat{a}^\dagger a}}\hat{S}^\dagger(\xi)\hat{D}^\dagger(\alpha), \tag{A.1}$$

where $\hat{D}(\alpha)$ and $\hat{S}(\xi)$ are respectively the displacement and the squeezing operators:

$$\hat{D}(\alpha) \equiv \exp\left[\alpha\hat{a}^\dagger - \alpha^*\hat{a}\right],$$
$$\hat{S}(\xi) \equiv \exp\left[(\xi^*\hat{a}\hat{a} - \xi\hat{a}^\dagger\hat{a}^\dagger)/2\right]. \tag{A.2}$$

The covariance matrix of the quadratures $\hat{X}$ and $\hat{P}$ for a generic state is defined as:

$$\boldsymbol{\sigma} \equiv \begin{pmatrix} \langle\hat{X}^2\rangle - \langle\hat{X}\rangle^2 & \langle\hat{X}\hat{P} + \hat{P}\hat{X}\rangle - \langle\hat{X}\rangle\langle\hat{P}\rangle \\ \langle\hat{X}\hat{P} + \hat{P}\hat{X}\rangle - \langle\hat{X}\rangle\langle\hat{P}\rangle & \langle\hat{P}^2\rangle - \langle\hat{P}\rangle^2 \end{pmatrix}. \tag{A.3}$$

For Gaussian states, every property can be expressed in terms of expectation values of the quadratures and their covariance matrix. In terms of the parameters $\alpha$, $\xi = re^{i\theta}$ and $N_{\text{th}}$, we

have:

$$\langle \hat{X} \rangle = \sqrt{2}\,\mathrm{Re}[\alpha], \quad \langle \hat{P} \rangle = \sqrt{2}\,\mathrm{Im}[\alpha], \tag{A.4}$$

$$\sigma_{11} = \left(\frac{1}{2} + N_{\mathrm{th}}\right)\Big(\cosh(2r) + \sinh(2r)\cos(\theta)\Big), \tag{A.5}$$

$$\sigma_{22} = \left(\frac{1}{2} + N_{\mathrm{th}}\right)\Big(\cosh(2r) - \sinh(2r)\cos(\theta)\Big), \tag{A.6}$$

$$\sigma_{12} = \sigma_{21} = \left(\frac{1}{2} + N_{\mathrm{th}}\right)\sinh(2r)\sin(\theta). \tag{A.7}$$

The von Neumann entropy of the photon in a Gaussian state reads:

$$S(\rho^G) = (N_{\mathrm{th}} + 1)\ln(N_{\mathrm{th}} + 1) - N_{\mathrm{th}}\ln N_{\mathrm{th}}. \tag{A.8}$$

We remark here that the origin of a finite entropy, i.e., $N_{\mathrm{th}} > 0$, is not generically guaranteed to be the entanglement with some other quantum system, unlike the closed cavity system in the main text, since it can also have a classical contribution. For example, a harmonic oscillator with frequency $\omega_{\mathrm{c}}$ and at inverse temperature $\beta$ is in a Gaussian state with $(\alpha, \xi, N_{\mathrm{th}}) = (0, 0, e^{-\beta\omega_{\mathrm{c}}})$.

Another definition for the Gaussian states is that their Wigner function:

$$W(x,p) = \frac{1}{\pi}\int dy\, e^{2ipy}\langle x + y|\hat{\rho}|x - y\rangle \tag{A.9}$$

is a Gaussian:

$$W(x,p) = \frac{1}{\pi}\exp\left(-\frac{1}{2}(x - x_0, p - p_0)\boldsymbol{\sigma}^{-1}(x - x_0, p - p_0)^T\right), \tag{A.10}$$

where $\boldsymbol{\sigma}$ it the covariance matrix, $x_0 = \langle \hat{X} \rangle$ and $p_0 = \langle \hat{P} \rangle$. We also recall that for a symmetrically ordered operator $\hat{O}(\hat{a}, \hat{a}^\dagger)$ such as the Peierls phase the Wigner function can be used to compute expectation values as averages over the phase space:

$$\langle \hat{O}(\hat{a}, \hat{a}^\dagger) \rangle = \int dx\, dp\, W(x,p)\, O\left(\frac{x + ip}{\sqrt{2}}, \frac{x - ip}{\sqrt{2}}\right). \tag{A.11}$$

In the main text, therefore, we need to perform just Gaussian integrals to arrive at Eq. (16).

## B  Photon mean-field in periodic boundary conditions

In this appendix, we expand on the case of periodic boundary condition without specifying the geometry. Using the same pseudo-spin representation defined in the text in momentum space we have that the light-matter Hamiltonian reads:

$$\hat{H} = \hat{H}_{\mathrm{c}} + \sum_{k,\alpha} H_k^\alpha(\hat{a}, \hat{a}^\dagger)\hat{\sigma}_k^\alpha, \tag{B.1}$$

where at each momentum sector $k$ we have:

$$H_k(\hat{a}, \hat{a}^\dagger) = -\Big(2t_0\cos(k)\cos(\hat{\Phi}/2), t_1 + t_2\cos(k), t_2\sin(k), 2t_0\sin(k)\sin(\hat{\Phi}/2)\Big). \tag{B.2}$$

Note that this representation is possible because the cavity mode has zero momentum in the direction of the ladder. Focusing on the thermodynamic limit and the matter state, we can work

in the PMF approximation and restrict ourselves to coherent states for the cavity $|\alpha_0\rangle$ with the identification of the mean-field parameter defined in the main text: $\phi = 4g\alpha_0/\sqrt{N}$ and $R = 1$. In this way the mean-field electronic Hamiltonian with periodic boundary conditions is

$$\hat{H}_{\mathrm{m}}^{\mathrm{PMF}} = \sum_{k,\alpha} h_k^\alpha \hat{\sigma}_k^\alpha . \tag{B.3}$$

The two bands are:

$$\epsilon_{k,a} = 2t_0 \cos(k)\cos(\phi) + (-1)^a \sqrt{t_1^2 + t_2^2 + 2t_1 t_2 \cos(k) + 4t_0^2 \sin^2(k)\sin^2(\phi)}, \tag{B.4}$$

with $a = 1, 2$. For the square ladder case discussed extensively in the main text with $t_1 = t_0 = 1$, the chemical potential is $\mu = 0$ at every $\phi$, and an indirect gap in the dispersion opens at $\phi = \frac{2}{3}\pi$. The minimization of the total energy as a function of $\phi$ then gives the PMF ground state.

## C  Perturbation theory

The Hamiltonian at $g = 0$ has a factorized ground state that reads:

$$|\Psi_0\rangle = \prod_{\frac{\pi}{3} < |k| < \frac{2\pi}{3}} \hat{\tilde{c}}_{1,k}^\dagger \prod_{|k| < \frac{\pi}{3}} \hat{\tilde{c}}_{1,k}^\dagger \hat{\tilde{c}}_{2,k}^\dagger |0_{\mathrm{m}}, 0_{\mathrm{c}}\rangle , \tag{C.1}$$

where $|0_{\mathrm{m}}, 0_{\mathrm{c}}\rangle$ is the state with zero electrons and photons, and $\hat{\tilde{c}}_{a,k}^\dagger$ is the creation operator that diagonalize the bare electronic Hamiltonian and $a = 1, 2$ the band index. Different points in momentum space can have $N_k = 0, 1, 2$ number of electrons and this is a conserved quantity. Starting from $|\Psi_0\rangle$ and the expansion of the light-matter interaction in Eq. (23) we can compute perturbative corrections at small $g$. In particular we have $H \simeq H_0 + V_g$ with:

$$\hat{V}_g = \frac{g}{\sqrt{L}} \sum_{k,\alpha} \hat{\sigma}_k^\alpha d_{p,k}^\alpha(\phi = 0) . \tag{C.2}$$

The only non-zero matrix matrix element at first order are those with a single direct particle-hole excitation for the matter and one photon in the cavity:

$$f_k = \langle PH_k, 1_{\mathrm{c}} | (\delta\hat{a} + \delta\hat{a}^\dagger) \sum_\alpha \hat{\sigma}_k^\alpha d_{p,k}^\alpha(\phi = 0) |\Psi_0\rangle = 2\sin(k)\theta\left(|k| - \frac{\pi}{3}\right)\theta\left(\frac{2\pi}{3} - |k|\right), \tag{C.3}$$

with $\theta(x)$ the Heaviside function. Summing over all momenta we arrive to the expression in the text for the ground state corrections:

$$\left|\Psi_{(1)}\right\rangle = |\Psi_0\rangle - \frac{g}{\sqrt{L}} \sum_k \frac{2f_k}{\omega_k + \omega_{\mathrm{c}}} |PH_k, 1_{\mathrm{c}}\rangle . \tag{C.4}$$

The second-order expansion involves also the $g^2$ contribution and populate also the two-photon sector of the cavity, needed for the squeezing of the mode.

## D  Details about DMRG simulations

For all the DMRG simulations performed here, the energy density difference between the final two DMRG sweeps has been kept below $10^{-8}$ to ensure convergence. In order to maintain

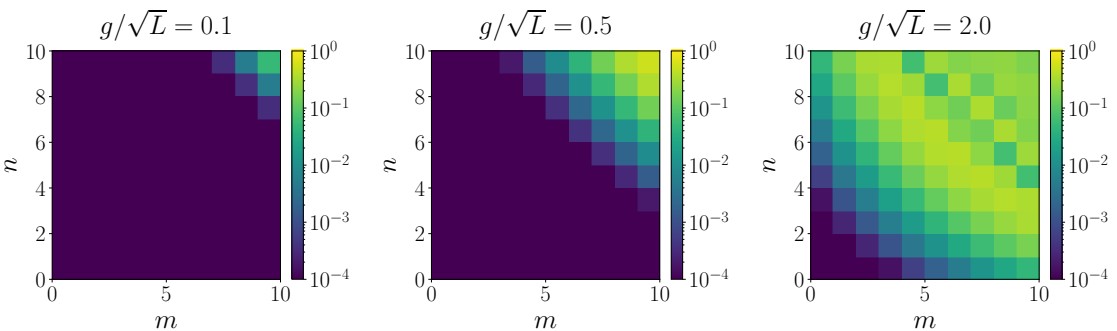

Figure 9: Absolute value of the difference $|D_{nm} - D_{nm}^{\text{exp}}|$ for three values of $g/\sqrt{L}$ where $D_{nm} = \langle n|\hat{D}(ig/\sqrt{L})|m\rangle$ and $D_{nm}^{\text{exp}}$ is obtained by exponentiating the finite matrix $X_{nm} = \langle n|\hat{X}|m\rangle$ in a truncated photon Hilbert space with $N_{\text{max}}^{\text{ph}} = 9$.

computational feasibility, the dimension of the photon Hilbert space has been truncated to a maximum photon number of $N_{\text{max}}^{\text{ph}} = 63$. The photon Hilbert space must be large enough to describe coherent states found in the photon condesed regime but also the strong squeezing. We have verified that this truncation level is sufficient to obtain converged results for all values of $g$ and system sizes up to $L = 76$.

In most of the presented figures, the bond dimension used for the MPS ansatz is $\chi = 600$, sufficient to achieve converged results for system sizes up to $L = 76$ with a tolerance of $10^{-8}$ on the energy density and a maximum truncation error of $10^{-6}$. These are worse case values which are found in the normal phase where the fermions are gapless and entangled with the cavity. However, to better capture the thermodynamic limit, we have also analyzed larger system sizes up to $L = 136$. For these system sizes, we increased the bond dimension to $\chi = 1000$ to converge most observables, except for the non-Gaussianity of the photon state. This observable has been found to be particularly sensitive to a combination of numerical parameters including the number of DMRG sweeps and the bond dimension. This problem of convergence is particularly pronounced for values of $g$ in the normal phase, where the entanglement in the system is higher. To account for this difficulty in the analysis, we have considered an empirical error of $10^{-5}$ in the data for Fig. 5.

We then also comment on the non-linear nature of the Peierls phase. This is represented in our code by using the exact matrix elements in the photon number basis $\{|n\rangle\}$ of the displacement operators $\hat{D}(ig/\sqrt{L})$ which reads [85]:

$$\langle n|\hat{D}(\alpha)|m\rangle = \sqrt{\frac{n!}{m!}}\,\alpha^{m-n}\exp\left(-\frac{|\alpha|^2}{2}\right)L_n^{(m-n)}\left(|\alpha|^2\right), \quad \text{for} \quad m \geq n, \tag{D.1}$$

with $L_n^{(m-n)}(x)$ a generalized Laguerre polynomial and for $m < n$ one can just take the complex conjugate since $\hat{D}$ is unitary. When one works at finite size or considers $g/\sqrt{L}$ fixed ("single-particle" strong coupling) the matrix elements of the displacement operator should be evaluated carefully in a truncated Hilbert space. For example the exponentiation of the matrix $ig\hat{X}/\sqrt{L}$ as $\langle n|\exp(ig\hat{X}/\sqrt{L})|m\rangle$ in a truncated Hilbert space does not exactly corresponds to $\langle n|\hat{D}(ig/\sqrt{L})|m\rangle$. To be more quantitative in Fig. 9 we plot the difference between the matrix elements obtained by exponentiating $ig\hat{X}/\sqrt{L}$ and the exact ones from Eq. (D.1) at a small photon Hilbert space cutoff $N_{\text{max}}^{\text{ph}} = 9$. This illustrates the necessity for a large photonic cut-off in the numerical simulations.

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
