# Peer review of "First-order photon condensation in magnetic cavities: A two-leg ladder model"

_SciPost Physics, doi:SciPost Phys. 15, 113 (2023)_

## Round 2 · Referee Report · Anonymous (Referee 1) · 2023-3-15

Strengths
- careful study of a Dicke-style model showing a first-order transition in equilibrium
- mean-field (+Gaussian fluctuations) compared to DMRG
- calculation of photon excitations
Weaknesses
- model most likely highly unrealistic
- some semantic issues (see report)
Report
This paper reports a careful study of a model of non-interacting fermions coupled to a single photon mode of a cavity. The coupling is "magnetic" and such systems have previously been shown to escape a famous no-go theorem which otherwise forbids equilibrium phase transitions in such models. The study is technically of high quality and includes a comparison to DMRG. A nice aspect is that the study also computes fluctuations beyond mean fields.
I have a number of questions/comments on the paper.
1) The paper uses the term "superradiant" 50 times throughout the paper including the title. While I understand the historical context of this term, nothing is "radiating" in the phases discussed in the paper.The formal justification is that the photon field develops a static expectation value but with the same justification one could identify any magnet (and in some gauges even systems with a static electric field, like a charge) as a "superradiant" state. The authors are fully aware of this issue and do identify their "superradiant" state correctly as a Condon phase (i.e., a phase with spontaneously created static magnetic field). Nevertheless, I think one should just avoid the term "superradiant"when absolutely nothing is radiating.
2) For me a main issue is (again this is something the authors explicitely mention in their paper) that the model and the investigated physical regime appear to be completely unrealistic. Current-current interactions are extremely small (suppressed by powers of the fine-structure constant) and I do not see how even a perfect cavity could help (there is no resonant enhancement anywhere and the induced magnetic fields are just static). Can it help to look for moiré systems or some suitably formed molecular networks?
3) The authors could, perhaps, add a discussion that mean field is exact for such type of models in the thermodynamic limit and that one can furthermore calculate fluctuation effects systematically in a 1/N expansion of precisely the form used by the authors. In this context it is surprising, that there still seems to be some mismatch of the 1/N results to DMRG extrapolated to large N. Does this have a trivial origin in 1/N effects arising from the difference of boundary conditions? Or is it a numerical issue?
4) The authors compare two models (square and triangular ladder) which both show the same behavior. A different behavior should occur if one goes for a system where an tiny flux already opens a gap at the Fermi energy. I expect this to happen in almost any 1d ladder model with a q=0 Dirac point (as, e.g., realized in metalic carbon nanotubes, but most likely also in some simple ladder systems). Did the authors consider this option?
In conclusion, while this is careful study by experts in the field, I am not convinced that it fulfills the acceptance criteria of scipost physics which asks for a breakthrough/groundbreaking result. Here one also has to take into account that the studied model is very unrealistic. The author seem to imply that the "groundbreaking" aspect of their study is the first-order nature of their transition but there are plenty of Dicke-style models with first order transitions.
Anonymous on 2023-03-30 [id 3524]
We thank the referee for the honest and factual report. We first reply on what is, in our opinion, the critical novelty element of our study. In the concluding remarks the referee writes:
<< The author seem to imply that the "groundbreaking" aspect of their study is the first-order nature of their transition but there are plenty of Dicke-style models with first order transitions.>>
Even though the spirit is similar to Dicke-style models we see two crucially defining differences in the presented ladder model. - The examples we have in mind of first order transitions (Refs [2,3]) have been proven, in a recently appeared work by some of us [1], to be artifacts of broken gauge invariance. Hence to the best of our knowledge, the present work is the first example of a first order photon condensation in a gauge invariant model.
-The nature of the matter degree of freedom is fundamentally different as our study deals with itinerant electrons and not localized emitters.
We now reply at the listed comments of the referee report:
<< 1) The paper uses the term "superradiant" 50 times throughout the paper including the title. While I understand the historical context of this term, nothing is "radiating" in the phases discussed in the paper.The formal justification is that the photon field develops a static expectation value but with the same justification one could identify any magnet (and in some gauges even systems with a static electric field, like a charge) as a "superradiant" state. The authors are fully aware of this issue and do identify their "superradiant" state correctly as a Condon phase (i.e., a phase with spontaneously created static magnetic field). Nevertheless, I think one should just avoid the term "superradiant"when absolutely nothing is radiating. >>
As the referee pointed out, the term “superradiant” has an historical motivation and only in the last years the term Condon phase has been reintroduced for such transitions. Given the terminology used in some recent related works [1] we would like to comply with the referee's suggestion and use the term “photon condensation” instead of “superradiant” transition. The title would change to “First-order photon condensation in magnetic cavities: A two-leg ladder model”.
<< 2) For me a main issue is (again this is something the authors explicitely mention in their paper) that the model and the investigated physical regime appear to be completely unrealistic. Current-current interactions are extremely small (suppressed by powers of the fine-structure constant) and I do not see how even a perfect cavity could help (there is no resonant enhancement anywhere and the induced magnetic fields are just static). >>
Even though we agree with the referee that our model is difficult to realize experimentally, as also clearly stated in the manuscript, we preferred to focus on more fundamental aspects which are well captured by the presented simple model. In particular we remark that it does not spoil basic conservation laws (gauge invariance) and focuses on first order photon condensation; investigating a previously unexplored mechanism for the appearance of a Condon phase in a system of itinerant electrons.
<< Can it help to look for moiré systems or some suitably formed molecular networks?>>
We note that Condon instabilities have already been explored in Moirè structures [4], but the focus has been on instabilities of the normal phase hence second order transitions. As a general statement, we agree that looking for a system with a huge unit cell area should, in principle, increase the effective magnetic coupling. However we kept ourselves from discussing the details of possible realization in physical systems as this would require a system-dependent analysis which is not the scope of this work. Moreover we want to stress that the quasi one dimensional nature of the model, even though far from possible 2 dimensional material realizations, allowed us to do a careful quasi-exact numerical treatment of the full problem with DMRG.
<< 3) The authors could, perhaps, add a discussion that mean field is exact for such type of models in the thermodynamic limit and that one can furthermore calculate fluctuation effects systematically in a 1/N expansion of precisely the form used by the authors. >>
We agree with the referee that photon mean field treatments for such models are exact when one is interested in thermodynamic properties, such as for scanning the matter phase diagram or calculating extensive matter observables. We might have not stressed it enough and we thank the referee for pointing this out. We plan to add a more explicit sentence in the discussion part and in the conclusions in a refined version of the manuscript. It is indeed one of the results of the present study to corroborate this idea with quasi-exact numerical calculations (DMRG).
<< In this context it is surprising, that there still seems to be some mismatch of the 1/N results to DMRG extrapolated to large N. Does this have a trivial origin in 1/N effects arising from the difference of boundary conditions? Or is it a numerical issue? >>
The very small mismatch (around 5x10^-3) in the extrapolated thermodynamic limit between DMRG (with opened boundary condition) and photon mean field + gaussian fluctuations (with periodic boundary conditions) is discussed relative to photonic observables which are converged within the DMRG numerics. However the extrapolation to the thermodynamic limit is done with a “naive” 1/L correction ansatz. Given the high degree of non-linearity of the coupling and the non additivity of the Hamiltonian we cannot exclude the presence of more subtle corrections which we cannot address with the presented data. We cannot also exclude the role of boundary conditions, even though we find it very unlikely that this should have an effect on the extrapolated values. As this point is not addressed in the present version of the manuscript we plan to add a more clear comment in this sense.
<< 4) The authors compare two models (square and triangular ladder) which both show the same behavior. A different behavior should occur if one goes for a system where an tiny flux already opens a gap at the Fermi energy. I expect this to happen in almost any 1d ladder model with a q=0 Dirac point (as, e.g., realized in metalic carbon nanotubes, but most likely also in some simple ladder systems). Did the authors consider this option? >>
We thank the referee for this insightful comment. We indeed tried different configurations of the ladder system, mainly by tuning the hopping parameters and filling of the ladders that in the presented results are instead fixed. We preferred to focus on two cases where the transition is first order to better convey the message but, as briefly mentioned in the conclusions, transitions of second order (small flux) are possible (e.g. t2=0 and t1>2*t0 at half filling is insulator to insulator, t2=0 and t1=t0 but quarter filling is metal to metal). Even though there are cases of the presented ladder models with q=0 dirac points (e.g. t0=1 t1=-t2>2), a finite magnetic flux as naively described in the manuscript does not open a gap but rather shifts the chemical potential. Of course we do not exclude that other ladder models where the q=0 Dirac point and/or the coupling to magnetic fields have a different nature could have a gap opening for small external magnetic fields but we think this interesting option should be addressed in a separate study.

---

## Round 3 · Referee Report · Anonymous (Referee 1) · 2023-5-15

Strengths

  • careful study of a model with long-ranged interactions showing a first-order transition in equilibrium
  • mean-field +Gaussian fluctuations compared to DMRG
  • calculation of photon excitations

Weaknesses

  • Model most likely highly unrealistic as current-current interactions are typically very small and very large magnetic fluxes are needed to realize the phase diagram.

Report

The authors have considered and answered all comments/questions of my previous report. Only the precise origin of discrepancies in 1/N still remains unclear (numerics or boundary conditions) but this is, perhaps, a detail.

It is not fully clear to me whether the study meets the acceptance criteria of scipost physics taking into accout that the model is unrealistic and that first-order transitions are in general not uncommon. Nevertheless, I support publication taking the overall quality of the paper into account.
  • validity: high
  • significance: good
  • originality: good
  • clarity: high
  • formatting: -
  • grammar: -

Author:  Zeno Bacciconi  on 2023-06-14  [id 3727]

(in reply to Report 1 on 2023-05-15)

We thank the Referee for their report. We would like to take the occasion to comment on the main weakness pointed out also by Referee 2 on the experimental relevance of the model. We recently become aware of an experiment (Ghirri et. al https://arxiv.org/abs/2302.00804) reporting ultra-strong magnetic coupling between magnonic excitations and a cavity mode confined on a superconducting surface. Although the nature of matter degrees of freedom in the experiment (spins) is different with respect to our work (itinerant electrons), we think that the experimental cavity set-up can be promising and is worth a more detailed investigation. We have added a sentence in the introduction to cite this work (now Ref. [47] in the manuscript) and slightly changed the discussion at the end of section 2 accordingly.

---

## Round 3 · Referee Report · Anonymous (Referee 2) · 2023-6-5

Strengths

A new proposal for a relatively bare-bones model that has a photon-condensation transition by avoiding no-go theorems.

Weaknesses

As mentioned before, the parameter values are probably not within reach under currently attainable experimental conditions.

Report

As this is a second round of reports and the manuscript has already been vetted regarding its validity, I can keep this short. Overall I do think that the manuscript meets the acceptance criteria for SciPost Physics. In particular, I vouch for interesting novel effect over being extremely realistic on the following grounds: The emergent field of cavity quantum materials is currently still very much in its exploratory stages. As such, even overly simplified models with parameters tuned to seemingly unrealistic values in order to achieve certain effects can have their merits in pushing forward the field by adding new ideas that can inspire follow-up studies. As such, I am for this paper.

Requested changes

Citations: - cavity superconductivity (refs. 8,9): an earlier theoretical proposal was https://www.science.org/doi/10.1126/sciadv.aau6969 - cavity ferroelectricity (ref. 10): here https://www.pnas.org/doi/10.1073/pnas.2105618118 should also be cited

  • validity: top
  • significance: top
  • originality: top
  • clarity: top
  • formatting: excellent
  • grammar: excellent

Author:  Zeno Bacciconi  on 2023-06-14  [id 3728]

(in reply to Report 2 on 2023-06-05)

We thank the Referee for their report and for the suggested references that we have added in the new version of the manuscript. We would like to take the occasion to comment on the main weakness pointed out also by Referee 1 on the experimental relevance of the model. We recently become aware of an experiment (Ghirri et. al https://arxiv.org/abs/2302.00804) reporting ultra-strong magnetic coupling between magnonic excitations and a cavity mode confined on a superconducting resonator. Although the nature of matter degrees of freedom in the experiment (spins) is different with respect to our work (itinerant electrons), we think that the experimental cavity set-up can be promising. We have added a sentence in the introduction to cite this work (now Ref. [47] in the manuscript) and slightly changed the discussion at the end of section 2 accordingly.

---

## Round 3 · Referee Report · Anonymous (Referee 3) · 2023-6-11

Strengths

It discusses the possibility of photonic condensation and a potential equilibrium superradiant phase transition in electronic systems. This transition is of first order, whereas typically in these systems, the transition is of second order.

Weaknesses

The paper should provide a better explanation as to why a phase transition exists in their case.

Report

The paper is very interesting. It is for two main reasons:

  • It discusses the possibility of photonic condensation and a potential equilibrium superradiant phase transition in electronic systems. This transition is of first order, whereas typically in these systems, the transition is of second order.

Therefore, the paper deserves consideration for publication. Before that happens, I would like the authors to discuss some of the following points:

  • Mean-field approximation: I would expect this approximation to be "exact" in the thermodynamic limit. Previous works by the authors have made this claim, e.g., their Refs. 25 and 29. By "exact," I mean that observables of both matter and light are exact in mean-field (even in the presence of light-matter entanglement, as in statistical mechanics). Is this the case? (It may be inferred from their study with system size.)

  • Relationship with no-go theorems: If I understand the paper correctly, the authors demonstrate that the light-matter system undergoes a quantum phase transition (QPT) as a function of the coupling parameter g. This type of transition is reminiscent of the equilibrium superradiant phase transition. As mentioned in the article, the literature is full of no-go theorems that prohibit this transition (some of these no-go theorems have been developed by the authors themselves). It seems that these no-go theorems do not apply here because the vector potential is not constant. However, the authors should explain why they do observe the transition while in other articles with similar systems (also with Peirls-type coupling), it is concluded that the transition cannot occur. Some of these articles are:

  • Ref 25 and its Section V, discussing the absence of a Superradiant Phase transition.

  • The Tight-binding model in their Ref 55.
  • The reference https://arxiv.org/abs/2302.08528 (Weber et al. "Cavity-renormalized quantum criticality in a honeycomb bilayer antiferromagnet") explicitly states (even in the abstract) that "While the position and universality class are not changed by a single cavity mode," and this fact is explained in the article by referring to the no-go theorems. By the way, I think this article is not cited.

In fact, in this manuscript, it is stated: "Our findings show that in the normal phase, when the photonic order parameter vanishes, collective coupling to a single cavity mode cannot change the properties of a thermodynamically large system."

However, in this system, the transition does occur, and therefore, the modification of matter properties happen.

In my opinion, the authors could use their model to understand the differences with other examples and the existence of the transition and modification of matter properties.

If this discussion is included, the paper deserves to be published.

  • validity: high
  • significance: good
  • originality: good
  • clarity: ok
  • formatting: good
  • grammar: excellent

Author:  Zeno Bacciconi  on 2023-06-14  [id 3729]

(in reply to Report 3 on 2023-06-11)

We thank the Referee for their report. We now reply to the questions raised by the Referee point by point:

<< * Mean-field approximation: I would expect this approximation to be "exact" in the thermodynamic limit. Previous works by the authors have made this claim, e.g., their Refs. 25 and 29. By "exact," I mean that observables of both matter and light are exact in mean-field (even in the presence of light-matter entanglement, as in statistical mechanics). Is this the case? (It may be inferred from their study with system size.)>>

Neglecting light-matter entanglement is justified if one is interested in phase diagram properties (in the thermodynamic limit) as the only sizable effect to the matter state can arise from a macroscopic classical photonic state which by definition does not carry light-matter entanglement. However the photon mean-field approximation is not exact when one is interested in observables of the cavity mode that are not extensive, e.g. the variance of the cavity quadrature X. In the present work we provided both numerical evidence with DMRG simulations and an analytical understanding with the treatment of gaussian fluctuations of the existence of light-matter entanglement in the ground state and its relevance for some cavity observables in the thermodynamic limit (Fig. 6).

<< * Relationship with no-go theorems: If I understand the paper correctly, the authors demonstrate that the light-matter system undergoes a quantum phase transition (QPT) as a function of the coupling parameter g. This type of transition is reminiscent of the equilibrium superradiant phase transition. As mentioned in the article, the literature is full of no-go theorems that prohibit this transition (some of these no-go theorems have been developed by the authors themselves). It seems that these no-go theorems do not apply here because the vector potential is not constant. However, the authors should explain why they do observe the transition while in other articles with similar systems (also with Peirls-type coupling), it is concluded that the transition cannot occur. Some of these articles are:

  • Ref 25 and its Section V, discussing the absence of a Superradiant Phase transition.
  • The Tight-binding model in their Ref 55.
  • The reference https://arxiv.org/abs/2302.08528 (Weber et al. "Cavity-renormalized quantum criticality in a honeycomb bilayer antiferromagnet") explicitly states (even in the abstract) that "While the position and universality class are not changed by a single cavity mode," and this fact is explained in the article by referring to the no-go theorems. By the way, I think this article is not cited. >>

The crucial difference with present no-go theorems and the references cited by the referee is that in these set-ups the coupling is purely electric ($\nabla \times A=0$) while in our work the cavity mode is magnetic ($\nabla \times A\neq0$). As this point seems to be unclear in the present version the manuscript, we have added a more explicit discussion in the introducion and in the conclusion.

<< In fact, in this manuscript, it is stated: "Our findings show that in the normal phase, when the photonic order parameter vanishes, collective coupling to a single cavity mode cannot change the properties of a thermodynamically large system." >>

We thank the Referee for pointing out this sentence as it indeed seems misleading. The message we want to convey is that, in the thermodynamic limit, the only way for single cavity mode to change the matter state in the collective strong coupling regime ($A_0\propto 1/\sqrt{N}$) is via a macroscopic coherent state. Hence, within the phases, quantum fluctuations of the cavity mode and light-matter entanglement only produce harmless $O(1)$ corrections. Differently at a quantum critical point, as discussed in the reference suggested by the referee, quantum fluctuations of the cavity mode will be in general important. Hence we have rephrased the sentence cited by the Referee in order to make the message more clear and added the suggested reference.

---

## Round 3 · Author Response

Dear Editor,

We thank you for handling our manuscript, and the Referee for the insightful report. We think we have addressed the Referee's main concern on lack of a ``groundbreaking" aspect. In particular, we believe that our manuscript meets Scipost Physics criteria for publication (1) and (3) for the corresponding two reasons:

  • Our work shows for the first time a first order photon condensation at equilibrium in a gauge invariant model. We want to stress that the few examples of first order transitions present in the literature have been shown to be artifact of model truncations.
  • The phenomenology readily understandable in the presented toy model should open new pathways in the quest for photon condensed phases in electronic systems.

Hence, we understand that the Editor can opt for the second option indicated in their recommendation, consulting at least another expert in the field in addition to the present Referee.

In our resubmitted manuscript, we have applied several changes motivated by the Referee's feedback. Below, as well as in the comment in reply to the Referee's report, please find a list of changes.

---

## Round 3 · List of Changes

- Use of photon condensation instead of superradiant transition in both title and text, some reference to the name equilibrium superradiant phase have been kept.
- Addition of Ref. [33] on Moire systems in the introduction (section 1).
- Addition of Refs. [42,43] on magnetic cavities in the introduction (section 1).
- Addition of Ref. [71] on 1/N corrections and a more explicit comment on mean-field treatments in the conclusions (section 5).
- Addition of Ref. [79] and a discussion on first order behaviors in magnetic systems in the conclusions (section 5).
- Addition of a comment on other first order superradiant transition propsed and addition of Refs. [77,78] in the conclusions (section 5).
- Discussion of thermodynamic limit comparison of DMRG and Gaussian fluctuations revised according to the reply to the referee report in the discussion (section 3.3).

---

## Round 4 · Referee Report · Anonymous (Referee 2) · 2023-6-14

Report

The paper is now ready for publication.

---

## Round 4 · Author Response

Dear Editor,

We thank you for handling our manuscript, and the Referees for their insightful reports. We have addressed the Referees comments and suggestions in the new version of the manuscript. Below, as well as in the comments in reply to the Referees' reports, please find a list of changes.

Yours sincerely,

The Authors

---

## Round 4 · List of Changes

• Addition of Ref. [47] and of a sentence in the introduction (section 1) that explains it. Slight modification of the last paragraph in section 2 where Ref [47] is discussed again in the context of experimental relevance of the considered model
  • Addition of Ref. [8,11] as suggested by Referee 2
  • Addition of a clearer discussion in the introduction (section 1, 4th paragraph) and in the conclusions (section 5, 1st paragraph) on the necessity of a magnetic coupling to evade no-go theorems.
  • Rephrasing of the sentence cited by Referee 3 (section 3.2, paragraph Magnetostatic instability) and addition of Ref. [75]

---

## Editorial Decision

published